# Target-agnostic identification of human antibodies to *Plasmodium falciparum* sexual forms reveals cross-stage recognition of glutamate-rich repeats

Axelle Amen[1,2†], Randy Yoo[3,4†], Amanda Fabra-García[5†], Judith Bolscher[6], William JR Stone[7], Isabelle Bally[1], Sebastián Dergan-Dylon[1], Iga Kucharska[3], Roos M de Jong[5], Marloes de Bruijni[6], Teun Bousema[5], C Richter King[8], Randall S MacGill[8], Robert W Sauerwein[6], Jean-Philippe Julien[3,4,9*‡], Pascal Poignard[1,2*‡], Matthijs M Jore[5*‡]

[1]CNRS, Université Grenoble Alpes, CEA, UMR5075, Institut de Biologie Structurale, Grenoble, France; [2]CHU Grenoble Alpes, Grenoble, France; [3]Program in Molecular Medicine, The Hospital for Sick Children Research Institute, Toronto, Canada; [4]Department of Biochemistry, University of Toronto, Toronto, Canada; [5]Department of Medical Microbiology, Radboud University Medical Center, Nijmegen, Netherlands; [6]TropIQ Health Sciences, Nijmegen, Netherlands; [7]Department of Immunology and Infection, London School of Hygiene and Tropical Medicine, London, United Kingdom; [8]Center for Vaccine Innovation and Access, PATH, Washington D.C., United States; [9]Department of Immunology, University of Toronto, Toronto, Canada

**\*For correspondence:**
jean-philippe.julien@sickkids.ca (J-PJ);
pascal.poignard@ibs.fr (PP);
matthijs.jore@radboudumc.nl (MMJ)

†These authors contributed equally to this work
‡These authors also contributed equally to this work

## eLife assessment

This study reports **important** results and new insights into humoral immune responses to *Plasmodium falciparum* sexual stage proteins. The experiments are based on the use of target-agnostic memory B cell sorting and screening approaches as well as several state-of-the-art technologies. The authors present **compelling** evidence that one antibody, B1E11K, is cross-reactive with multiple proteins containing glutamate-rich repeats through homotypic interactions, a process similar to what has been observed for *Plasmodium* circumsporozoite protein repeat-directed antibodies.

**Abstract** Circulating sexual stages of *Plasmodium falciparum* (*Pf*) can be transmitted from humans to mosquitoes, thereby furthering the spread of malaria in the population. It is well established that antibodies can efficiently block parasite transmission. In search for naturally acquired antibodies targets on sexual stages, we established an efficient method for target-agnostic single B cell activation followed by high-throughput selection of human monoclonal antibodies (mAbs) reactive to sexual stages of *Pf* in the form of gametes and gametocyte extracts. We isolated mAbs reactive against a range of *Pf* proteins including well-established targets Pfs48/45 and Pfs230. One mAb, B1E11K, was cross-reactive to various proteins containing glutamate-rich repetitive elements expressed at different stages of the parasite life cycle. A crystal structure of two B1E11K Fab domains in complex with its main antigen, RESA, expressed on asexual blood stages, showed binding of B1E11K to a repeating epitope motif in a head-to-head conformation engaging in affinity-matured homotypic interactions. Thus, this mode of recognition of *Pf* proteins, previously described only for Pf circumsporozoite protein (PfCSP), extends to other repeats expressed across various stages. The findings augment our understanding of immune-pathogen interactions to

repeating elements of the *Plasmodium* parasite proteome and underscore the potential of the novel mAb identification method used to provide new insights into the natural humoral immune response against *Pf*.

## Introduction

The eradication of malaria remains a global health priority. In 2021, 247 million people were diagnosed with malaria with over 619,000 people succumbing to the disease (*WHO, 2021*). Malaria is caused by *Plasmodium* parasites: unicellular eukaryotic protozoans that transmit to human hosts through an *Anopheles* mosquito vector. Although insecticide-treated nets and various antimalarial compounds have aided in controlling the spread and combating severe clinical complications of the disease, mosquito and parasite strains resistant to such interventions have emerged (*Wicht et al., 2020*). Thus, there is an urgent need to develop other technologies, such as vaccines, to help combat the spread of malaria, and eventually, contribute to the eradication of the disease.

The development of a highly efficacious malaria vaccine has been challenging, owing to the complex life cycle of malaria-causing parasites. The life cycle of *Plasmodium* spp. can be categorized into three distinct stages: the pre-erythrocytic, asexual blood stage, and sexual stages. Each step features the parasite undergoing large-scale morphological changes accompanied by modifications in proteomic expression profiles – with several proteins being exclusively expressed in a single stage (*Florens et al., 2002*). Therefore, vaccine-mediated humoral responses must be robust enough to generate sufficiently high-quality antibody responses to eliminate the majority of parasites at the stage the vaccine targets. As a result, one approach to vaccine development has focused on targeting 'bottlenecks' in the parasite's life cycle where the number of parasites is low (*Cowman et al., 2016*; *Rosenberg, 2008*).

The pre-erythrocytic stage – following the transmission of sporozoites to humans by a bite from an infected mosquito – is a bottleneck targeted by the only two malaria vaccines recommended by the World Health Organization, RTS,S/AS01 and R21/Matrix-M (*Laurens, 2020*; *Datoo et al., 2024*). The vaccines are based on the circumsporozoite protein (CSP) – an essential protein expressed at high density on the surface of the parasite during the pre-erythrocytic stage (*Cerami et al., 1992*; *Frevert et al., 1993*; *Ménard et al., 1997*). In its central domain, the protein contains various four amino acid repeat motifs – the most predominant of those being NANP repeats. Protective antibodies elicited against CSP primarily target this immunodominant repeat region. A large proportion of these antibodies engage in homotypic interactions which are characterized by direct interactions between two antibodies' variable regions when bound to adjacent repetitive epitopes (*Kucharska et al., 2022a*; *Kucharska et al., 2022b*; *Imkeller et al., 2018*; *Martin et al., 2023b*; *Pholcharee et al., 2021*; *Oyen et al., 2018*; *Martin et al., 2023a*; *Murugan et al., 2020*; *Tripathi et al., 2023*). Such interactions have been demonstrated to augment B cell activation and contribute to shaping the humoral response to CSP (*Imkeller et al., 2018*). Although these first-generation malaria vaccines only induce short-lived efficacy in a subset of the at-risk population (*RTS,S Clinical Trials Partnership, 2015*), they will undoubtedly assist in lowering the overall incidence of malaria in young infants with clear initial impact (*Wadman, 2023*). Modeling suggests that for improvements toward eradication of the disease, combining multiple types of interventions that target different stages of the *Plasmodium* life cycle may strongly increase the efficacy of weaker interventions (*Golumbeanu et al., 2022*).

Another bottleneck in the parasite's life cycle occurs during its sexual stage, which takes place in the mosquito vector. This stage begins when intraerythrocytic *Plasmodium* gametocytes, which are capable of eliciting antibody responses through clearance in the spleen (*Stone et al., 2016*), are taken up through a mosquito blood meal. Once inside the mosquito midgut, gametocytes emerge from erythrocytes and mature into gametes which then undergo fertilization ultimately resulting in the generation of sporozoites that can go on to infect the next human host. Vaccines that aim to target this life cycle bottleneck are known as transmission-blocking vaccines (TBVs) (*Nikolaeva et al., 2015*; *Duffy, 2021*). The goal of TBVs is to elicit antibodies that target sexual stage antigens to block the reproduction of the parasite in the mosquito and thus onward transmission to humans. Current TBV development efforts are primarily focused on two antigens, Pfs230 and Pfs48/45, as they are the targets of the most potent transmission-blocking antibodies identified to date (*Roeffen et al., 2001*; *Ivanochko et al., 2023*; *Fabra-García et al., 2023*; *Kundu et al., 2018*). Indeed, individuals with sera

enriched in antibodies that target Pfs230 and Pfs48/45 were found to have high transmission-reducing activity (TRA) (*Stone et al., 2018*). Protective responses to Pfs230 and Pfs48/45 are well characterized at a molecular and structural level (*Ivanochko et al., 2023*; *Fabra-García et al., 2023*; *Kundu et al., 2018*; *Coelho et al., 2021*; *Tang et al., 2023*; *Ko et al., 2022*; *Lennartz et al., 2018*; *Singh et al., 2020*). However, sera depleted in antibodies targeting the pro-domain and domain 1 of Pfs230 and domains 2 and 3 of Pfs48/45 can retain TRA while maintaining the ability to recognize the surface of parasites lacking Pfs48/45 and Pfs230 surface expression (*Stone et al., 2018*). This highlights the importance of expanding our understanding of antibody responses to other sexual stage antigens.

We designed a workflow for the isolation of antibodies to sexual stage-specific antigens from a donor who was repeatedly exposed to malaria parasites. Our efforts yielded a panel of 14 monoclonal antibodies (mAbs) targeting Pfs230 and Pfs48/45 as well as other unidentified proteins, some with TRA. One mAb exhibited cross-reactivity to multiple antigens present at various stages of the parasite's life cycle, by targeting glutamate-rich repeats and engaging in homotypic interactions. This latest result underscores a pivotal role of repetitive elements in shaping the humoral response to *Plasmodium falciparum* (*Pf*).

## Results

### Agnostic memory B cell (MBC) sorting and activation identifies potential *Pf* sexual stage protein-specific mAbs

We selected donor A, a 69-year-old Dutch expatriate who resided in Central Africa for approximately 30 years and whose serum was shown to strongly reduce *Pf* transmission, to isolate sexual stage-specific mAbs. We previously demonstrated the serum of this donor, donor A, largely retained its TRA when depleted of antibodies directed against the main transmission-blocking epitopes of Pfs48/45 and *Pf*s230 (*Stone et al., 2018*), suggesting the presence of antibodies targeting other epitopes on these two proteins or directed at other proteins also involved in transmission.

PBMCs from the donor were thawed and a total of 1496 IgG+ memory B cells were sorted in 384-well plates (*Figure 1*) (*Figure 1—figure supplement 1A*). After activation, single-cell culture supernatants potentially containing secreted IgGs were screened in a high-throughput 384-well ELISA for their reactivity against a crude *Pf* gamete lysate (*Figure 1—figure supplement 1B*). A subset of supernatants was also screened against gametocyte lysate (*Figure 1—figure supplement 1C*). In total,

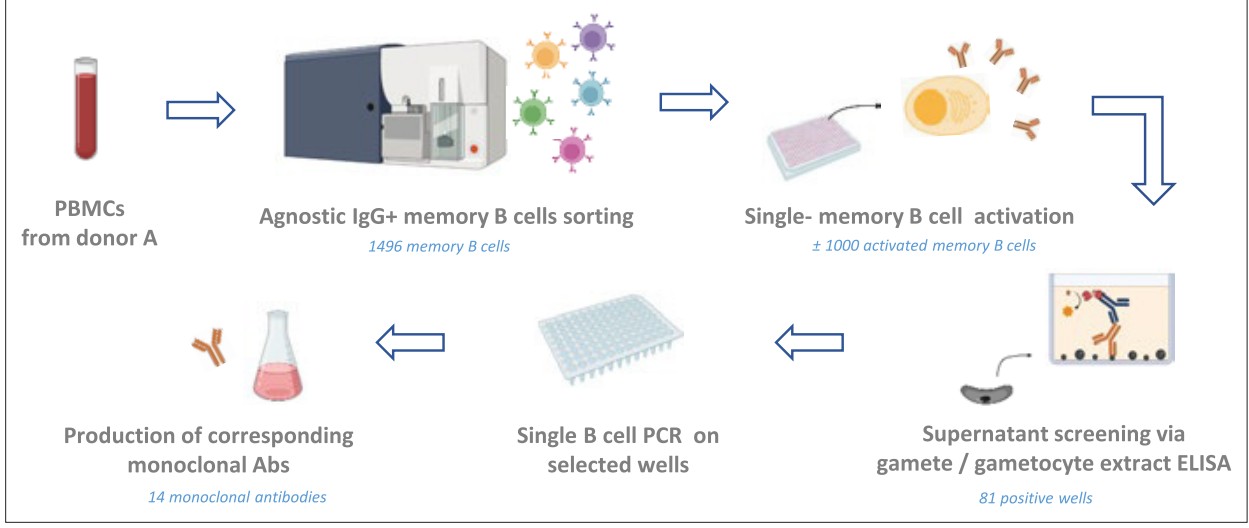

**Figure 1.** General workflow. IgG+ memory B cells from donor A were sorted individually regardless of their specificity, at one cell per well. Cells were further cultured in activation medium with CD40L-expressing feeder cells and cytokines to induce antibody secretion. Supernatants were tested for antibody binding to the sexual stage of the parasite through screening using a gamete extract ELISA. Memory B cells from wells displaying reactivity were selected for Ig genes amplification, followed by cloning and production of the corresponding antibody. Figure was created with BioRender.

The online version of this article includes the following figure supplement(s) for figure 1:

**Figure supplement 1.** Memory B cell (MBC) sorting and cell culture supernatant screening.

supernatants from 84 wells reacted with gamete and/or gametocyte lysate proteins, representing 5.6% of the total memory B cells. Of the 21 supernatants that were screened against both gamete and gametocyte lysates, six recognized both, while nine appeared to recognize exclusively gamete proteins, and six exclusively gametocyte proteins.

To isolate the corresponding mAbs, single B cell lysates from the 84 ELISA-positive wells were subjected to single B cell reverse transcriptase PCR (RT-PCR) for amplification of immunoglobulin variable genes. We obtained and cloned heavy and light chain sequences for 11 out of 84 wells. For three wells we obtained a kappa light chain sequence and for five wells a lambda light chain sequence. For three wells we obtained both a lambda and kappa light chain sequence suggesting that either both chains were present in a single B cell or that two B cells were present in the well. For all 14 wells we retrieved a single heavy chain sequence. Following amplification and cloning, 14 mAbs were expressed as full human IgG1s (*Supplementary files 1 and 2*).

## Isolated mAbs exhibit distinct patterns of recognition of gamete surface proteins

The 14 mAbs were first tested for binding to *Pf* sexual stage surface antigens in a surface immuno-fluorescence assay (SIFA) using wild-type female gametes (*Figure 2A*). The mAbs were also tested for binding to Pfs48/45 knockout female gametes, which lack surface-bound Pfs48/45 and Pfs230 (*Eksi et al., 2006*; *Stone et al., 2018*) Seven mAbs exhibited binding to approximately half or more of gametes when tested at a concentration of 100 µg/mL. Among these, four mAbs, B1C5K, B1C5L, B2C10L, and B2E9L, recognized wild-type gametes with high scores (>68%) even at concentrations as low as 1 µg/mL. The binding of the B1C5K, B1C5L, B2C10L, and B2E9L mAbs strongly decreased when using gametes that lacked surface-expressed Pfs48/45 and Pfs230, indicating that these four mAbs likely targeted one of these two antigens. Three other mAbs, B2D10L, B1C8L, and B1E7K, displayed a similar recognition profile, albeit with notably smaller percentages of labeled wild-type gametes, particularly at the lower concentrations tested. This suggested a potential low-affinity recognition of either Pfs48/45 or Pfs230 for these latter three mAbs.

Six of the remaining seven mAbs, B1C8K, B1D3L, B1D3K, B1F9K, B1C3L, and B2F7L, exhibited very weak or no binding to gametes. For B1C8K, this showed that the light chain (kappa) did not correspond to the antibody that was originally selected in the screening process as the lambda version (B1C8L) exhibited strong binding. As for the other mAbs, the results indicated that they may be specific for proteins not expressed or only poorly expressed at the gamete surface.

Finally, one mAb, B1E11K, exhibited a distinctive gamete surface binding profile, recognizing only a fraction (approximately a third to a fifth) of the wild-type and Pfs48/45 knockout gametes across all tested concentrations, suggesting potential binding to non-Pfs48/45 and Pfs230 proteins.

## Isolated mAbs have varying TRAs and recognize different *Pf* sexual stage proteins

We were interested in investigating potential TRA for all identified mAbs. To do this, a standard membrane feeding assay standard membrane feeding assay (SMFA) was conducted in the presence of the isolated mAbs, revealing a range of TRAs (*Figure 2B*; *Figure 2—source data 1*). Overall, seven mAbs were confirmed to strongly reduce transmission (TRA>80%) when tested at 500 µg/ml: B1C5K, B1C5L, B2C10L, B2E9L, B1C8L, B1D3L, and B1F9K. Of those, two mAbs, B1C8L and B2C10L, retained >50% TRA at a lower concentration (100 µg/ml). Notably, despite not showing gamete surface recognition, B1F9K and B1D3L displayed TRA – although only at high concentrations. Conversely, three mAbs recognizing the gamete surface, B2D10L, B1E7K, and B1E11K, showed no activity in SMFA (Standard membrane feeding assay).

To gain deeper insight into the specificity of the various isolated mAbs, regardless of their TRA, we conducted further characterization via western blot analyses using gametocyte extracts (*Figure 2C*). The B1C5K and B1C5L mAbs recognized a protein with a molecular weight matching that of Pfs48/45. This result was consistent with the findings from the gamete surface binding experiment in which these mAbs recognized wild-type gametes but not Pfs48/45 knockout gametes, providing further validation of the specificity of the B1C5K and B1C5L mAbs in targeting Pfs48/45. The B2C10L mAb displayed pattern of recognition that corresponded to Pfs230, similar to the anti-Pfs230 control mAb RUPA-96 (*Ivanochko et al., 2023*). Once again, these results were in agreement with the findings from

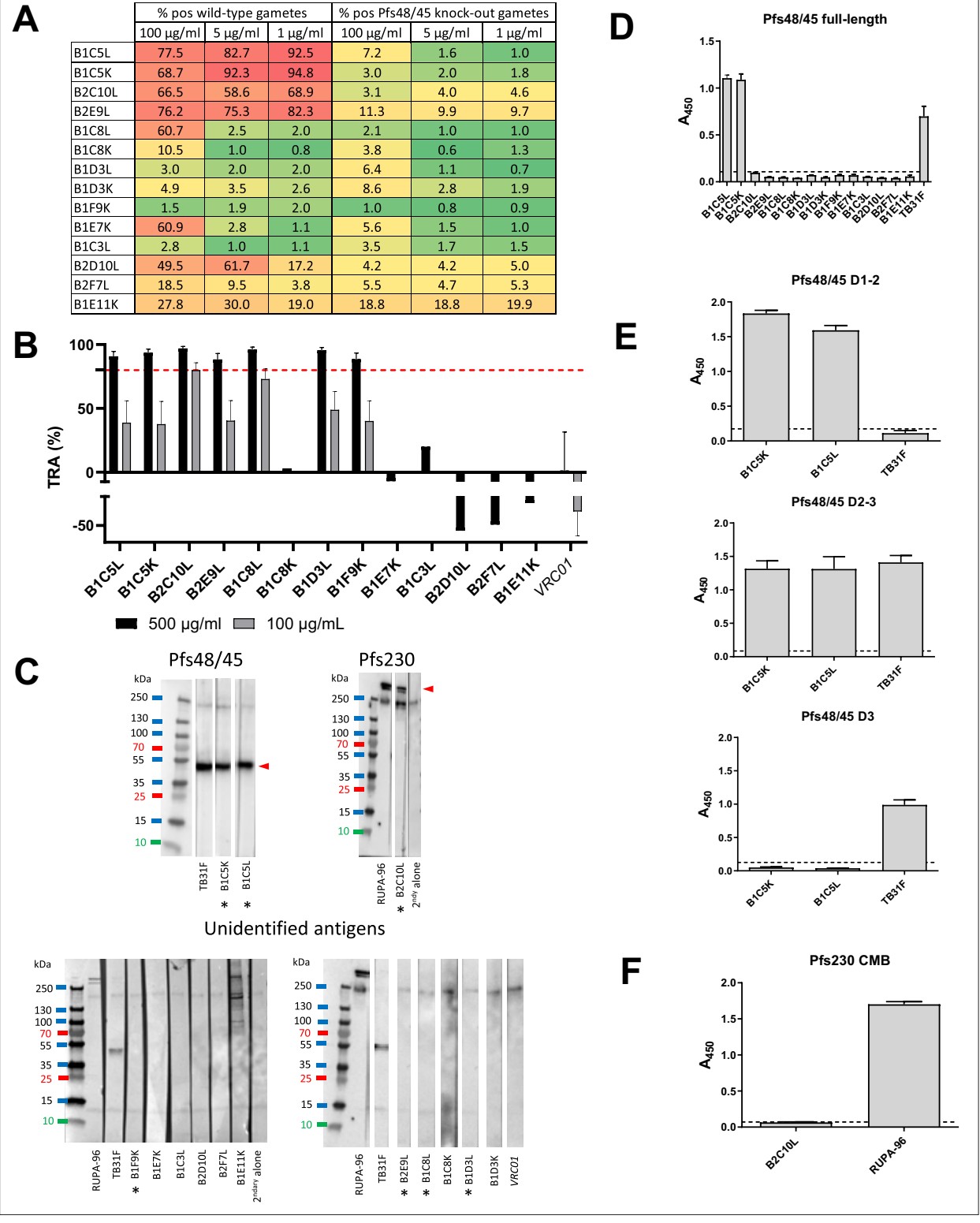

**Figure 2.** Characterization of the panel of isolated monoclonal antibodies (mAbs). (**A**) Percentage positive wild-type gametes and Pfs48/45 knockout (KO) gametes that also lack surface-bound Pfs230 in surface immunofluorescence assay, in a heatmap format (graded color scale: red for high percentage of binding, green for low percentage of binding). The experiment was performed in duplicate and three different mAb concentrations were tested (100 µg/ml, 5 µg/ml, and 1 µg/ml). (**B**) Transmission-reducing activity (TRA) of the mAb panel in standard membrane feeding assay (SMFA). For mAbs with >80% TRA at 500 µg/ml, experiments were run in duplicates and bars are estimates of the mean and error bars represent the 95% confidence intervals. mAbs with >80% TRA at 500 µg/ml were also tested at 100 µg/ml. Oocyst count data of the SMFA (Standard membrane feeding assay)

*Figure 2 continued on next page*

*Figure 2 continued*

experiments can be found in *Figure 2—source data 1*. (**C**) Reactivity of the mAb panel against gametocyte extract in western blot, in non-reducing conditions. Antibodies are classified depending on the antigen recognized: Pfs48/45, Pfs230, or no antigen identified. TB31F is an anti-Pfs48/45 mAb, RUPA-96 is an anti-Pfs230 mAb, and VRC01 is an anti-HIV mAb (negative control). Pfs48/45 and Pfs230 bands are indicated with a red arrow, antibodies with >80% TRA at 500 µg/ml are indicated with an asterisk (*). (**D**) Reactivity of the mAb panel at 30 µg/ml against full-length Pfs48/45 in ELISA. (**E**) B1C5K and B1C5L binding to various Pfs48/45 domains in ELISA, at 10 µg/ml. (**F**) B2C10L binding to Pfs230CMB domain in ELISA, at 10 µg/ml. Values in (D-F) are means from three technical replicates and error bars represent standard deviation. mAbs were considered positive when the absorbance was higher than the mean absorbance plus three standard deviations of seven negative mAbs, indicated by dashed lines.

The online version of this article includes the following source data for figure 2:

**Source data 1.** Raw standard membrane feeding assay (SMFA) data.

**Source data 2.** Original western blots.

the gamete surface binding experiment confirming the specificity of the B2C10L mAb in targeting Pfs230. The B1E11K mAb also appeared to bind to Pfs230 on the western blot. However, in contrast to the RUPA-96 mAb, B1E11K only recognized the higher molecular band corresponding to Pfs230, suggesting exclusive recognition of the unprocessed form of this protein (*Brooks and Williamson, 2000*). Interestingly, besides Pfs230, B1E11K also recognized several unidentified proteins ranging from 70 kDa and 250 kDa with various intensities. These findings were consistent with the gamete binding assay that showed recognition of gametes lacking Pfs48/45 and Pfs230, suggesting potential recognition of proteins other than Pfs48/45 and Pfs230. Finally, all the other mAbs showed no clear binding to any protein from the gametocyte extract on the western blot. In the case of those demonstrating binding to gamete surfaces, B2E9L, B2D10L, B1C8L, B1E7K, this may be attributed to their recognition of conformational epitopes that are lost during western blotting preparation, or possibly of specific recognition of proteins expressed in gametes but not gametocytes.

The recognition of Pfs48/45 by the B1C5K and B1C5L mAbs was subsequently confirmed through an ELISA using full-length recombinant Pfs48/45 (*Figure 2D*). Notably, none of the other mAbs of the panel displayed binding to this Pfs48/45 recombinant protein construct. To further pinpoint the domain targeted by B1C5K and B1C5L, an ELISA was performed using constructs corresponding to domains 1–2, 2–3, and 3 revealing that these two mAbs targeted domain 2 of Pfs48/45 (*Figure 2E*). This is in agreement with prior work indicating mAbs to domain 2 of Pfs48/45 are generally mAbs with low potency (*Fabra-García et al., 2023*).

The combined findings above strongly pointed to Pfs230 as the target of mAb B2C10L. We thus tested this mAb in ELISA for binding to Pfs230CMB, a construct containing Pfs230 domain 1 and part of the pro-domain (*Farrance et al., 2011*). No reactivity was observed (*Figure 2F*) suggesting that B2C10L may recognize other Pfs230 domains than the one tested, or recognize epitopes not properly displayed in the construct used.

In summary, our target-agnostic mAb isolation approach successfully identified mAbs against *Pf* sexual stage proteins, some of which exhibited TRA and some of which target Pfs48/45 or Pfs230. However, given that the mAbs isolated in this study showed substantially lower TRA than mAbs identified previously (*Kundu et al., 2018*; *Coelho et al., 2021*), we elected not to investigate them further. Instead, we were intrigued by the binding properties of B1E11K, which showed cross-reactivity with various *Pf* proteins, including Pfs230. Such cross-reactivity has been shown as a hallmark of the human antibody response to *Pf* and explored at the serum level but to our knowledge has never been studied at the mAb level (*Hou et al., 2020*; *Raghavan et al., 2023*). Thus, we rationalized a more detailed molecular characterization of this mAb may provide insights into this relatively unexplored phenomenon.

## The B1E11K mAb cross-reacts to distinct sexual and asexual stage *Pf* proteins containing glutamate-rich repeats

First, to ensure the ability of B1E11K to recognize different proteins in western blotting experiments was not due to polyreactivity, the mAb was tested in ELISA against a panel of human proteins, single-stranded DNA (ssDNA) and lipopolysaccharide (LPS). The 4E10 mAb, a well-known anti-HIV gp41 polyreactive mAb (*Cardoso et al., 2005*), was used as a positive control. The B1E11K mAb did not bind any of the antigens on the panel at any significant level, even at a high 50 µg/ml concentration, and therefore polyreactivity was ruled out (*Figure 3—figure supplement 1A*).

To identify antigens recognized by B1E11K, immunoprecipitation experiments were conducted using gametocyte extract. Proteins of different molecular weights were specifically detected in the B1E11K immunoprecipitate but not when using the anti-HIV control mAb (*Figure 3—figure supplement 1B*). Mass spectrometry analysis of the corresponding gel slices revealed recognition of Pfs230, confirming the western blot results.

The specificity of B1E11K was further tested using a protein microarray featuring recombinant proteins corresponding to putative antigens expressed at the sexual stage as well as proteins expressed at different stages of the *Pf* life cycle (*Stone et al., 2018*). The results showed that B1E11K exhibited high level reactivity (>8-fold higher than the negative control, minimum signal intensity rank 15th of 943 array targets) against several antigens, some expressed at the sexual stage (i.e. Pf11.1), others at the asexual stage (i.e. LSA3, RESA, RESA3) (*Figure 3A*; *Figure 3—source data 1*). Analysis of the primary amino acid sequence of the antigens recognized in the array suggested homology in several cases, based on the presence of glutamate-rich regions (*Figure 3—figure supplement 2*). To analyze the numerous repeated motifs contained in these proteins, we used the RADAR (Rapid Automatic Detection and Alignment of Repeats) software (*Heger and Holm, 2000*). Although B1E11K recognition of Pfs230 fragments on the array was lower than our cutoff for further analysis (3.4-fold higher than the negative control, maximum signal intensity rank 30th of 943 array targets), its sequence was also analyzed using RADAR due to its recognition by B1E11K in the immunoprecipitation experiments (*Figure 3—figure supplements 3 and 4*). The analysis showed Pfs230 and several of the proteins recognized by B1E11K on the array contained diverse patterns of glutamate-rich repeats of different lengths and compositions. Among these proteins, Pfs230, Pf11.1, RESA, RESA3, and LSA3 presented the most similar glutamate repeats, following an 'EE-XX-EE' pattern (*Figure 3—figure supplements 3 and 4*). Pfs230 contains adjacent EE-VG-EE repeats which are located in the domain of the protein which is cleaved upon gametocyte egress from erythrocytes (*Williamson et al., 1996*). RESA and RESA3 contain 20 and 9 EE-NV-EE overlapping repeats at the C-terminus of the protein, respectively. LSA3 contains two overlapping EE-NV-EE repeats. Finally, 221 non-adjacent EE-LV-EE repeats span the whole Pf11.1 megadalton protein.

To verify the recognition of the aforementioned proteins, a western blot was performed with recombinant forms of RESA, RESA3, LSA3, and of a Pf11.1 domain (*de Jong, 2023*; *Figure 3B*). Domain 1 of Pfs230, which does not contain the EE-VG-EE repeats, was also included. The results confirmed the binding of B1E11K to all the proteins tested except for Pfs230D1, as expected. Overall, the data showed that the B1E11K mAb recognizes various *Pf* proteins from different stages, all containing glutamate-rich repeats.

To validate the B1E11K mAb specifically targets glutamate-rich repeats, we synthesized five biotinylated peptides derived from the various repeats found in the proteins identified above to test for binding in sandwich ELISA experiments (*Figure 3C*). Overall, the B1E11K mAb bound to all peptides (*Figure 3D*). However, it exhibited a higher specificity for the RESA-derived peptides with an $EC_{50}$ at least 100 times greater compared to $EC_{50}$ values obtained with the other glutamate-rich peptides. Altogether, this suggests the main antigenic targets of B1E11K are RESA and RESA3, which contain the EENV repeats.

Since B1E11K bound to RESA-based peptides the strongest, we synthesized shorter RESA peptides for a more precise determination of the B1E11K minimal sequence epitope (*Figure 4A*). When tested on the RESA peptide panel, B1E11K mAb binding to RESA P2 (16AA) and RESA 14AA, 12AA, and 10AA was similar, all exhibiting close $EC_{50}$ values (*Figure 4B*; *Figure 4—figure supplement 1*). No binding was observed for the 8AA RESA peptide, suggesting the 10AA peptide contained the minimal epitope. We hypothesized the similar $EC_{50}$ values may be a result of avidity effects and thus, we performed the same experiment with recombinant B1E11K Fab (*Figure 4C*). Although B1E11K Fab bound both RESA P2 (16AA) and RESA 14AA peptides with comparable strength of binding, both RESA 12AA and RESA 10AA peptides displayed comparatively poorer $EC_{50}$ values with the RESA 10AA peptide displaying the lowest detectable binding strength. The binding affinity and kinetics of the interaction was also determined through biolayer interferometry (BLI). We performed experiments using the minimal sequence required for binding determined through ELISA (RESA 10AA peptide and the RESA P2 peptide [16AA]). When immobilizing the peptide to the sensors, an approximately six-fold difference in affinity between the 10AA peptide ($K_D$ = 484 nM) (*Figure 4D*) and the P2 peptide (16AA) ($K_D$ = 74 nM) (*Figure 4E*) was observed.

**A**

| Peptide ID | Corresponding protein | Peptide position in the protein *Peptide length* | Normalised score | Corrected score |
|---|---|---|---|---|
| PF3D7_0220000 | Liver Stage Antigen 3 | 67 to 822 *756* | 6.253495 | 59988.5 |
| PF3D7_0309100 | OMD protein | Full length *187* | 6.188559 | 57356 |
| PF3D7_1149200 | ring infected erythrocyte surface antigen 3 (RESA3) | 65 to 587 *523* | 6.193454 | 54235.5 |
| PF3D7_0102200 | ring infected erythrocyte surface antigen (RESA) | 568 to 1085 (end) *518* | 6.044558 | 53268 |
| PF3D7_1127500 | Protein disulfide isomerase | Full length *433* | 5.913398 | 44843.5 |
| PF3D7_1036300 | Duffy binding like merozoite surface protein 2 | 423 to 762 *340* | 5.433248 | 32032.5 |
| PF3D7_0411700 | conserved plasmodium protein | 138 | 5.111879 | 25673.5 |
| PF3D7_1038400 | Pf 11.1 | 2980 to 3637 *658* | 4.763914 | 21787 |
| PF3D7_1149200 | ring infected erythrocyte surface antigen 3 | 570 to 1090 *521* | 4.723332 | 20946.5 |
| PF3D7_0804500 | uncharacterized protein | 215 | 4.610963 | 19703 |
| PF3D7_1038400 | Pf 11.1 | 8737 to 9663 (end) *827* | 3.986108 | 12776.5 |
| PF3D7_1038400 | Pf 11.1 | 7795 to 8628 *778* | 3.757038 | 11117.5 |
| PF3D7_0909000 | uncharacterized protein | 528 | 3.689088 | 10267 |
| PF3D7_1038400 | Pf 11.1 | 103 to 1024 *922* | 3.242688 | 7017 |
| PF3D7_0522400 | uncharacterized protein | 934 | 3.164308 | 6643.5 |

**Figure 3.** B1E11K binds repeat peptides. (**A**) B1E11K binding to recombinant fragments of *Plasmodium falciparum* (*Pf*) proteins displayed on a microarray. (**B**) B1E11K binding to several recombinant proteins in western blot, in non-reducing conditions. (**C**) Sequences of the peptides tested for binding. Peptides were N-terminally linked to a biotin moiety using aminohexanoyl (Ahx) spacers. (**D**) B1E11K binding in ELISA to a panel of peptides.

*Figure 3 continued on next page*

*Figure 3 continued*

The online version of this article includes the following source data and figure supplement(s) for figure 3:

**Source data 1.** Raw microarray data.

**Source data 2.** Original western blot.

**Figure supplement 1.** Further characterization of B1E11K.

**Figure supplement 2.** Sequences of the recombinant protein fragments in the microarray.

**Figure supplement 3.** Glutamic acid-rich repeats in RESA (**A**), RESA3 (**B**), LSA3 (**C**), and Pfs230 (**D**).

**Figure supplement 4.** Glutamic acid-rich repeats in Pf11.1.

## Four EENV repeats permit two B1E11K Fabs to bind

Given the repetitive nature of the antigenic targets of B1E11K and differences in binding events captured in our BLI experiments, we hypothesized that more than one B1E11K Fab could potentially bind to the longer, RESA P2 (16AA) peptide. Thus, we performed isothermal titration calorimetry (ITC) using the same two RESA peptides as in the BLI experiments to determine the binding stoichiometries. We observed when titrating the B1E11K Fab into RESA 10AA, a binding stoichiometry of N=1.0 ± 0.2

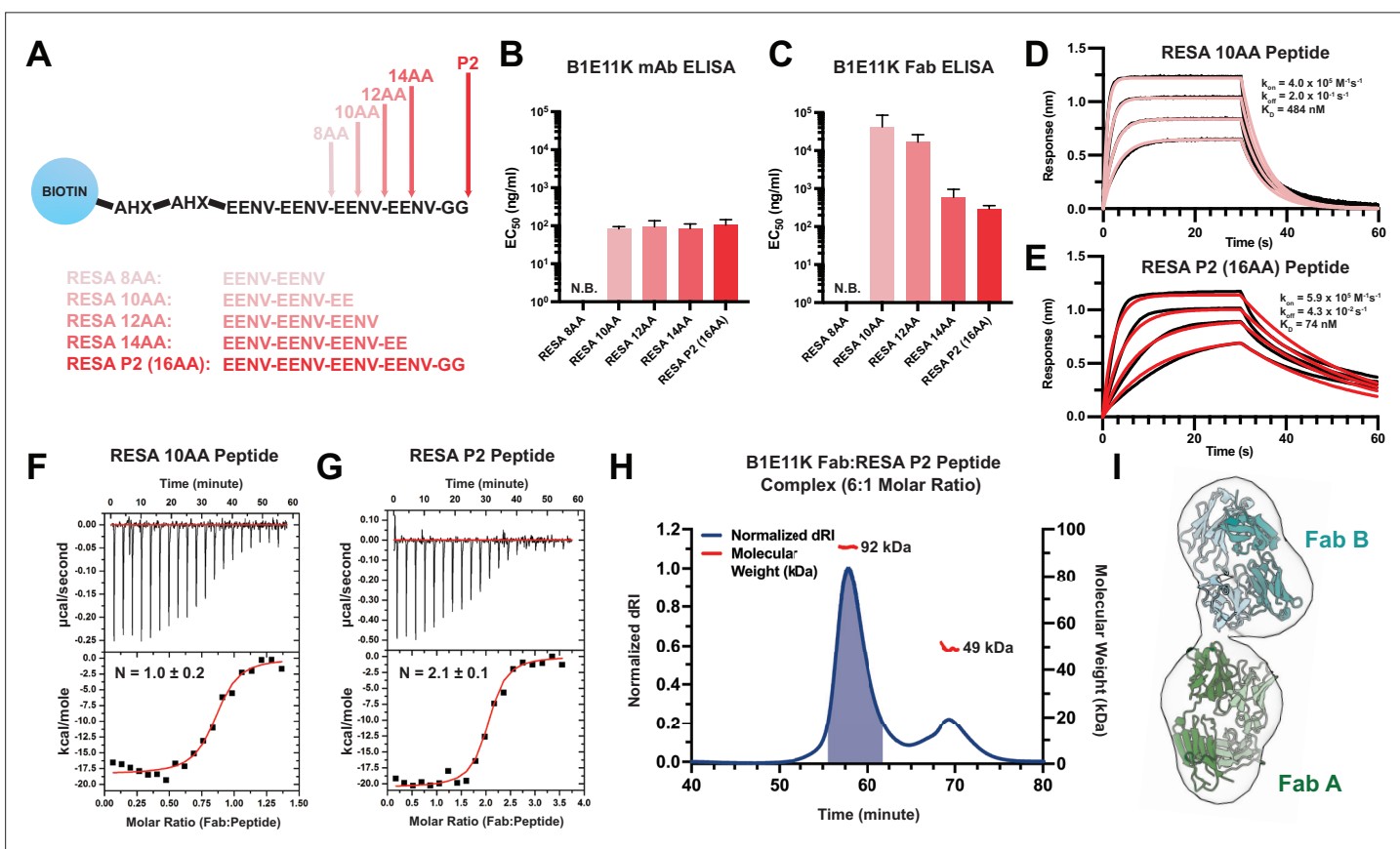

**Figure 4.** Binding characteristics of RESA peptides to B1E11K. (**A**) Various peptides based on the EENV repeat region were designed and conjugated to a biotin-AHX-AHX moiety (AHX = ε-aminocaproic acid). EC$_{50}$ values obtained from ELISA experiments utilizing various EENV repeat peptides with (**B**) B1E11K mAb or (**C**) B1E11K Fab. Error bars represent standard deviation. Biolayer interferometry experiments utilizing immobilized (**D**) RESA 10AA peptide or (**E**) RESA P2 (16AA) peptide dipped into B1E11K Fab. Representative isothermal titration calorimetry experiments in which B1E11K Fab was injected into (**F**) RESA 10AA peptide or (**G**) RESA P2 (16AA) peptide. (**H**) Size-exclusion chromatography coupled with multi-angle light scattering (SEC-MALS) of a solution of B1E11K Fab incubated with RESA P2 (16AA) peptide in a 6:1 molar ratio. The predicted molecular weight of the B1E11K Fab and RESA P2 peptide are 46.9 kDa and 2.5 kDa, respectively. The shaded region indicates the fractions collected used for negative-stain electron microscopy (nsEM). (**I**) An nsEM map reconstruction which permits the fitting of two B1E11K Fabs (Fab A and Fab B).

The online version of this article includes the following figure supplement(s) for figure 4:

**Figure supplement 1.** BE11K ELISA binding curves to RESA peptides.

**Table 1.** Isothermal titration calorimetry (ITC) thermodynamics and binding affinity of B1E11K Fab to RESA peptides.

|  | RESA 10AA peptide (n=2) | RESA P2 peptide (n=3) |
|---|---|---|
| N | 1.0±0.2 | 2.1±0.1 |
| $K_D$ (nM) | 78±12 | 73±21 |
| ΔG (kcal/mole) | –9.7±0.3 | –9.8±0.3 |
| ΔH (kcal/mole) | –18.3±0.1 | –20.5±0.1 |
| –TΔS (kcal/mole) | 8.6±0.2 | 10.7±0.3 |

Error reported as standard deviation.

(*Figure 4F*; *Table 1*). When using the RESA P2 (16AA) peptide, a stoichiometry of N=2.1 ± 0.1 was observed (*Figure 4G*; *Table 1*). The determined binding affinity from our ITC experiments (*Table 1*) differed from our BLI experiments (*Figure 4D and E*), which can occur when measuring antibody-peptide interactions (*Kratochvil et al., 2021*). Regardless, our data all trend toward the same finding in which a stronger binding affinity is observed toward the longer RESA P2 (16AA) peptide.

To further corroborate our binding stoichiometry findings, we performed size-exclusion chromatography coupled with multi-angle light scattering (SEC-MALS) to determine the molecular weight of the 2:1 Fab:peptide complex (*Figure 4H*). We incubated a molar excess Fab:peptide (6:1) sample to saturate all B1E11K Fab binding sites present on the RESA peptide to obtain a solution containing the putative complex and excess monomeric Fab. The resulting chromatogram revealed two species eluted from the column. The molecular weight of the heavier species was in line with what would be expected from a 2:1 Fab:peptide complex (92 kDa) in which the mass determined fell within the range of experimental error. A negative-stain electron microscopy (nsEM) map reconstruction of the 2:1 Fab:peptide complex recovered from the SEC-MALS experiment (*Figure 4H*) permitted the fitting of two Fab molecules, further supporting the 2:1 binding model (*Figure 4I*).

## B1E11K binds EENV repeats in a head-to-head conformation leveraging homotypic interactions

To obtain a full structural understanding of the observed repeat cross-reactivity and selectivity for RESA exhibited by B1E11K, we solved a 2.6 Å crystal structure of the B1E11K:RESA P2 (16AA) peptide complex (*Figure 5A*; *Table 2*; *Supplementary file 3*). The electron density at the binding interface is unambiguous and included density for the entirety of the repeat region of the peptide (*Figure 5—figure supplement 1*). Looking at the binding interface between the two Fabs and peptide reveals the structural basis for cross-reactivity (*Table 1*; *Figure 5—figure supplement 2*; *Supplementary file 4*). The paratope of B1E11K is highly enriched in arginine and histidine residues giving rise to a highly electropositive groove (*Figure 5B–D*). These residues form a plethora of salt-bridging interactions with the glutamate residue side chains of RESA repeats (*Figure 5E*). These interactions are supplemented by hydrogen bonding interactions of backbone serine and glycine residues of the B1E11K paratope as well as a hydrogen bond involving W33 found in the heavy chain of Fab B. Multiple hydrogen bonding interactions are made with B1E11K through the side chains of the asparagine residues of RESA repeats (EENV) (*Figure 5F*) that would not exist in the context of binding to the repeats of Pf11.1 (EELV or EEVIP or EEFIP or EEVVP) or Pfs230 (EEVG) (*Figure 3C*), as these residues lack side chains that can form hydrogen bonds. This likely leads to the observed higher specificity of B1E11K for RESA repeats demonstrated in our ELISA experiments (*Figure 3D*).

Additionally, the crystal structure of the antibody-antigen complex revealed the presence of homotypic antibody-antibody contacts through two interfaces surrounding the repeat peptide binding groove (*Figure 5G*; *Figure 5—figure supplement 3*; *Supplementary file 5*). The first interface features a salt-bridging network involving D60 of the Fab B kappa chain forming two salt bridges with R52 and H52A of the Fab A HCDR2 (*Figure 5H*). Additionally, R54 of the Fab A HCDR2 forms two hydrogen bonds and a salt bridge with the Fab B kappa chain side chains of S55 and E56. Finally, Y58 of the Fab B HCDR2 forms a hydrogen bond with the side chain of S53 of the Fab A HCDR2. The second interface is less extensive featuring two hydrogen bonds between Y32 and S31 of the Fab B HCDR1 and the Fab A KCDR1 T29 and R27, respectively (*Figure 5I*).

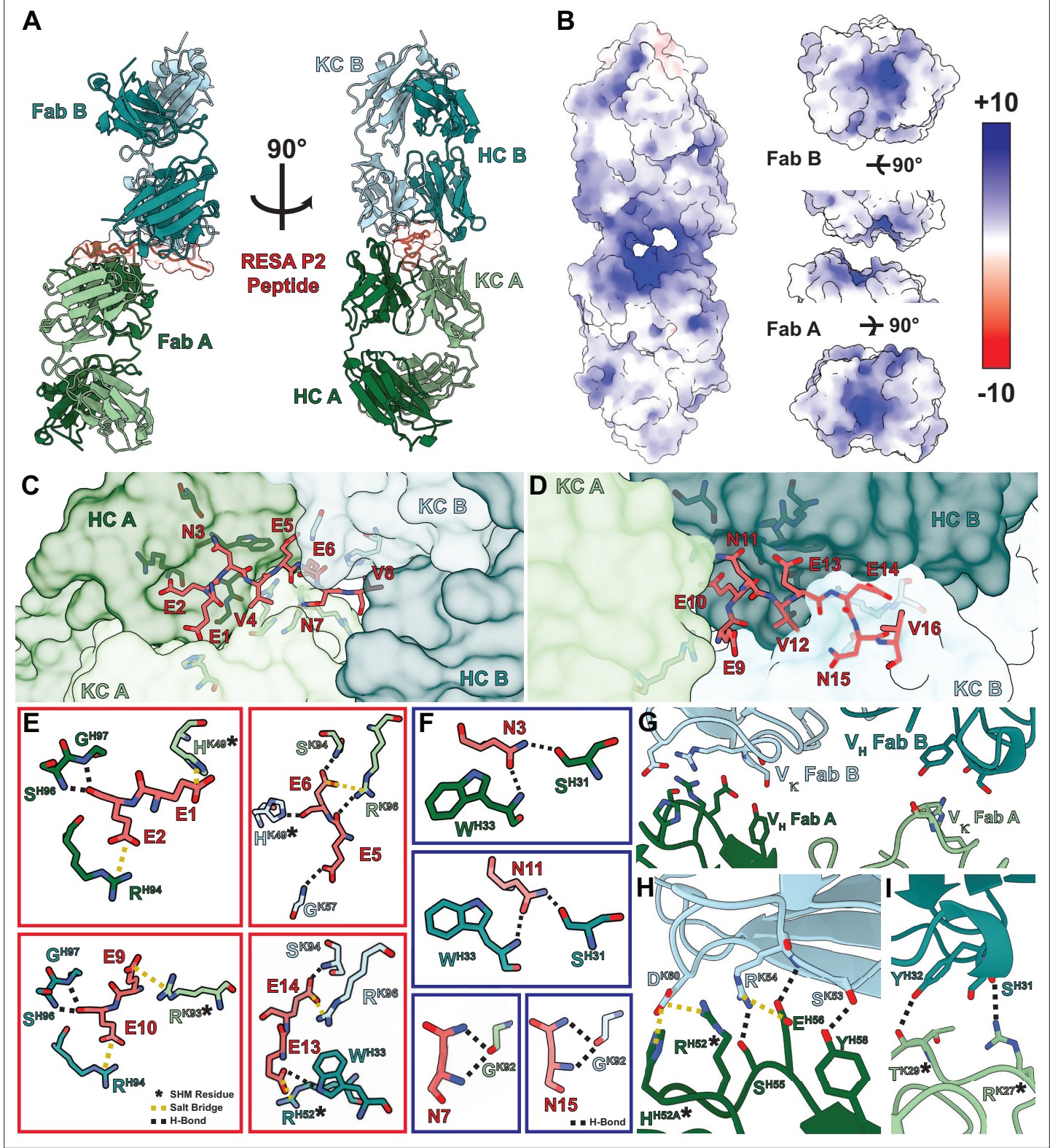

**Figure 5.** Structure of the B1E11K Fab and RESA P2 (16AA) peptide complex. (**A**) The overall architecture of the B1E11K:RESA P2 (16AA) peptide complex. (**B**) The electrostatic potential of the surface of the B1E11K Fabs. Fab residues involved in electrostatic interactions with (**C**) residues 1–8 and (**D**) 9–16 of the RESA P2 peptide are shown as sticks. (**E**) Electrostatic interactions occurring with glutamate residues of the RESA P2 (16AA) peptide. Residues that have undergone somatic hypermutation (SHM) are marked with an asterisk. Salt bridges are shown as dashed yellow lines and hydrogen

*Figure 5 continued on next page*

*Figure 5 continued*

bonds as dashed black lines. (**F**) Hydrogen bonding interactions through the asparagine residues of the RESA P2 (16AA) peptide are shown as black dashed lines. (**G**) Variable heavy (V$_H$) and variable kappa (V$\kappa$) residues involved in homotypic interactions are shown as sticks. (**H**) The first interaction interface and (**I**) second interface. Residues that have undergone SHM are marked with an asterisk. Electrostatic interactions are presented as dashed lines and colored as done previously.

The online version of this article includes the following figure supplement(s) for figure 5:

**Figure supplement 1.** Composite omit maps of residues involved in inter-chain interactions.

**Figure supplement 2.** Buried surface area plots of B1E11K Fabs and RESA P2 (16AA) peptide interactions.

**Figure supplement 3.** Buried surface area plots of B1E11K Fabs of residues buried at the homotypic interaction interface.

**Figure supplement 4.** IgBLAST of B1E11K heavy chain and light chain.

Analysis of the B1E11K sequences with IgBLAST (*Ye et al., 2013*) reveals that the B1E11K heavy and light chain have high similarity to the IGHV3-7 and IGKV3-20 germline sequences (*Figure 5—figure supplement 4*). This analysis also indicates multiple residues involved in the homotypic interaction interface have undergone somatic hypermutation. Residues of the CDR2 heavy chain of Fab A, R52 and H52A, and kappa chain CDR1 residues of Fab B, T29 and R27, form various electrostatic and van der Waals interactions which are mutated from the inferred germline sequences (*Figure 5F and G*; *Figure 5—figure supplements 3 and 4*). In summary, our biophysical and structural characterization revealed the basis of cross-reactivity and specificity to RESA repeats (EENV) through a binding interface highly electrostatic in nature, featuring affinity-matured homotypic interactions between adjacent antibody molecules when in its antigen-bound state.

## Discussion

Here, we introduce an innovative strategy that circumvents the use of recombinant proteins, to explore humoral immunity to *Pf* sexual proteins. We used target-agnostic MBC sorting and activation, followed by screening to assess reactivity against *Pf* gamete lysate and, for a fraction of the sorted cells, gametocyte lysate. The approach enabled the identification of a panel of mAbs targeting diverse *Pf* proteins, including some that exhibit TRA. The total number of isolated antibodies was relatively low due to the limited number of cells used. However, the identified B cells accounted for 5.6% of the total number, which suggest that there is a relatively high proportion of B cells specific for gamete proteins in the memory compartment of this donor, considering not all B cells were activated (typically 70–80% activation in our experiments). Furthermore, despite screening with a parasite extract containing a mixture of intracellular and surface proteins, half of the mAbs displayed binding to the surface of gametes, and/or exhibited TRA. This suggests antibody responses to surface proteins and proteins involved in transmission were high in this particular donor, potentially explaining the potent TRA observed with the serum.

Among the seven mAbs exhibiting TRA, three were found to recognize the well-defined TRA targets Pfs48/45 and Pfs230. The B1C5L and B1C5K mAbs were shown to recognize domain 2 of Pfs48/45 and exhibited moderate potency, as previously described for antibodies with such specificity (*Fabra-García et al., 2023*). These two mAbs were isolated from the same well and shared the same heavy chain; their similar characteristics thus suggest that their binding is primarily mediated by the heavy chain. Furthermore, a mAb identical to B1C5K was recently isolated from the same donor using a single B cell selection approach with recombinant Pfs48/45 (*Fabra-García et al., 2023*). The B2C10L mAb was shown to recognize Pfs230 in gamete binding assays and western blot, but failed to bind the pro- and D1 domains of Pfs230 in ELISA. This confirms TRA may be mediated through binding to domains other than pro-D1, the current main vaccine candidate (*Miura et al., 2022*), as previously observed for rodent mAbs that were generated against native Pfs230 (*Simons et al., 2023*; *Inklaar et al., 2023*; *de Jong et al., 2021*).

The remaining four mAbs exhibiting TRA did not clearly demonstrate recognition of either Pfs230 or Pfs48/45. Among those mAbs, B2E9L, and to a lesser extent B1C8L, showed recognition of wild-type gametes but not gametes that lacked surface-bound Pfs230 and Pfs48/45. However, western blot did not identify any protein targeted by these two mAbs and they did not bind to Pfs48/45 in ELISA. Therefore, we hypothesize these two mAbs target a protein associated with the Pfs48/45-Pfs230

**Table 2.** Crystallography statistics.

| Crystal | B1E11K:RESA P2 (16AA) peptide |
|---|---|
| Beamline | APS-23-ID-B |
| Wavelength (Å) | 1.0332 |
| Space group | C 2 2 2₁ |
| Cell dimensions | |
| a, b, c (Å) | 78.7, 186.3, 131.5 |
| α, β, γ (°) | 90, 90, 90 |
| Resolution (Å)* | 40.0–2.56 (2.65–2.56) |
| No. molecules in ASU | 1 |
| No. of observations | 236,070 (23,834) |
| No. unique observations | 31,520 (3067) |
| Multiplicity | 7.5 (7.8) |
| $R_{merge}$ (%)[†] | 14.5 (222.8) |
| $R_{pim}$ (%)[‡] | 5.7 (85.5) |
| $<I/\sigma I>$ | 10.8 (1.0) |
| $CC_{1/2}$ (%) | 99.7 (36.5) |
| Completeness (%) | 99.8 (98.5) |
| Refinement statistics | |
| Reflections used in refinement | 31,512 |
| Reflections used in R-free | 1575 |
| Non-hydrogen atoms | 6781 |
| Macromolecule | 6627 |
| Water | 130 |
| Heteroatom | 24 |
| $R_{work}$[§]/$R_{free}$[¶] (%) | 21.5/24.5 |
| Rms deviations from ideality | |
| Bond lengths (Å) | 0.002 |
| Bond angle (°) | 0.48 |
| Ramachandran plot | |
| Favored regions (%) | 96.0 |
| Allowed regions (%) | 3.8 |
| Ramachandran outliers (%) | 0.2 |
| B-factors (Å²) | |
| Wilson B-factor | 65.5 |
| Average B-factors | 89.8 |
| Average macromolecule | 90.3 |
| Average heteroatom | 83.1 |
| Average water molecule | 61.0 |

*Values in parentheses refer to the highest resolution bin.

[†]$R_{merge} = \Sigma_{hkl} \Sigma_i \mid I_{hkl,i} - <I_{hkl}> \mid / \Sigma_{hkl} <I_{hkl}>$.

[‡]$R_{pim} = \Sigma_{hkl} [1/(N-1)]^{1/2} \Sigma i \mid I_{hkl,i} - <I_{hkl}> \mid / \Sigma_{hkl} <I_{hkl}>$.

[§]$R_{work} = (\Sigma \mid |Fo| - |Fc| \mid) / (\Sigma \mid |F_o| \mid)$ – for all data except as indicated in footnote ¶.

[¶]5% of data were used for the $R_{free}$ calculation.

complex (*Simon et al., 2016*). An alternative explanation for the antigenicity of B2E9L and B1C8L is that these two mAbs may target a Pfs230 conformational epitope that is not represented in western blot assays. The last two mAbs that exhibited TRA, B1D3L and B1F9K, displayed no reactivity with the gamete surface, neither with wild-type nor gametes that lacked surface-bound Pfs48/45 and Pfs230, and did not show any recognition in western blot. In the case of B1D3L, the selection during screening was based on recognition of the gametocyte lysate while testing on the gamete extract was negative. This mAb may possibly target a protein only expressed on gametocytes (indicating that the epitope might be conformational and not properly displayed in western blotting with gametocyte extract). Regarding B1F9K, it is somewhat surprising that this mAb, which was originally selected based on positivity in the gamete extract ELISA, did not display reactivity in SIFA while still exhibiting TRA. Further exploration is needed to understand this apparent discrepancy. Overall, these latter mAbs, which do not recognize well-defined TRA targets, demonstrated lower potency in the SMFA (Standard membrane feeding assay) compared to some of the best characterized mAbs with such activity. Nevertheless, they could still be of strong interest in defining potential novel TRA targets, and further investigations are needed.

Seven of the 14 mAbs isolated did not exhibit TRA. Of those, four exhibited some level of binding to gamete surfaces: B1E11K, B2D10L, B1E7K, and B2F7L (albeit very weakly). This may suggest either the recognition of proteins involved in transmission but with an insufficient affinity to exert a significant effect, recognition of non-functional epitopes in proteins that play a role in transmission, or the recognition of proteins unrelated to the transmission process altogether.

B1E11K recognized various proteins from both the Pf sexual stage and asexual stages, all containing glutamate-rich repeats. Repetitive regions rich in glutamate residues have been previously found to be highly immunogenic in malaria-experienced individuals. A study investigating antibody responses against asexual stage antigens of *Pf* associated with erythrocyte invasion using sera from individuals from various cohorts found that the repetitive regions rich in glutamate residues within these antigens were predominantly recognized (*Hou et al., 2020*). Another investigation into sera from individuals from Uganda corroborated this finding (*Raghavan et al., 2023*). Raghavan et al. noted the antibodies that target these repeats may be potentially cross-reactive but emphasized that such a claim could only be demonstrated by direct investigations into mAbs. To our knowledge, only four mAbs that target glutamate-rich repeats have been described in which their epitopes have been determined. Of those mAbs, three were obtained following mouse immunization, and only one was of human origin. The murine mAbs 1A1 (*Feng et al., 1993*) and 1E10 (*Scherf et al., 1992*) recognize Pf11.1-derived repeat peptides ([PEE(L/V)VEEV(I/V)]$_2$); the murine mAb 9B11 (*Masuda et al., 1986*) is able to bind to a peptide containing four EENV repeats of RESA; and finally the human mAb 33G2 (*Udomsangpetch et al., 1986*) is specific for a peptide repeat sequence found in Ag332 (VTEEI) (*Ahlborg et al., 1991*). Despite all targeting linear epitopes containing tandem glutamate residues (EE), only 33G2 appeared to exhibit cross-reactivity (*Udomsangpetch et al., 1989*) and none had been structurally characterized. Thus, our structure provides critical insights into how glutamate-rich-repeat targeting antibodies from immune individuals can cross-react with various *Pf* proteins expressed at different life cycle stages.

A most revealing observation from our structure is the presence of affinity-matured antibody-antibody homotypic interactions in the context of recognizing repetitive tandem glutamate residues present across the *Pf* proteome. The finding that B1E11K targets a repetitive epitope while engaging in affinity-matured homotypic interactions is similar to how antibodies elicited against repetitive elements of CSP can also bind through homotypic interactions (*Imkeller et al., 2018*; *Pholcharee et al., 2021*; *Oyen et al., 2018*; *Martin et al., 2023b*; *Martin et al., 2023a*; *Tripathi et al., 2023*; *Kucharska et al., 2022a*; *Kucharska et al., 2022b*; *Murugan et al., 2020*). We have previously shown that B cells expressing B cell receptors (BCRs) interacting via homotypic interactions activate more robustly in comparison to B cells that have mutated BCRs that disrupt this interaction (*Imkeller et al., 2018*). This strong B cell activation, presumably mediated through the cross-linking of multiple BCRs at the B cell surface (*Clutterbuck et al., 2012*), has been suggested to limit affinity maturation in germinal centers, potentially due to early exit of B cells, favoring the elicitation of short-term low-affinity antibodies to CSP over durable high-affinity responses, thus leading to suboptimal protective responses (*Wahl and Wardemann, 2022*). However, this phenomenon may potentially be altered with the development of cross-reactive responses against repeats of slightly different content (*Murugan*

et al., 2020; Ludwig et al., 2023; Thai et al., 2023). Whether such insights extend to other anti-repeats antibody responses in general and anti-glutamate-rich repeats in particular remains largely unexplored.

Our observation of the recognition of RESA glutamate repeats by the B1E11K mAb through homo-typic interactions tends to confirm a generalizable property of B cell responses to repetitive antigens where antibodies can bind in close proximity. Here, we observed that B1E11K mAb exhibits a fair degree of somatic hypermutation and a relatively high affinity for RESA and cross-reactivity to other antigens. This finding provides further credence to the proposition that high-affinity-matured anti-bodies to repeats can be elicited when cross-reacting to motifs of slightly different content, in the present case derived from antigens expressed at different stages of the Pf life cycle. Nonetheless, cross-binding to repeats-sharing proteins from different stages, as demonstrated with the B1E11K mAb, could also represent another mechanism by which repeated motifs may impact protective responses. Indeed, antibodies elicited by one protein with repeats may hinder subsequent potential protective responses to cross-recognized proteins expressed later in the parasite life cycle through antibody feedback mechanisms such as epitope masking (Raghavan et al., 2023; Chatterjee et al., 2021; Chatterjee and Cockburn, 2021; McNamara et al., 2020).

Ultimately, understanding the dynamics of how the immune system responds to repetitive elements could be critical for the future rational design of malaria vaccines. A desired characteristic for next-generation malaria vaccines will be the ability to elicit antibodies that can inhibit at multiple stages of the parasite's life cycle to prevent infection, reduce clinical manifestations, and lower the spread of the disease (Nahrendorf et al., 2015; Julien and Wardemann, 2019). This could be accomplished by designing a multi-stage malaria vaccine that displays antigens expressed at various points of the parasite's life cycle. Utilizing glutamate-rich repeats in such a design may present benefits as a single antigen could potentially give rise to sera which contain antibodies that can lower both the clinical burden (asexual stage-targeting antibodies) and transmission of the disease (sexual stage-targeting antibodies). Indeed, antibodies elicited against the repetitive elements described here have been demonstrated to be associated with a lower incidence of disease (Riley et al., 1991; Petersen et al., 1990; Berzins et al., 1991) as well as disrupt the maturation of sexual stage parasites (Feng et al., 1993) in in vitro assays – although we note that mAb B1E11K isolated and characterized in this study did not show TRA.

As such, future work will be necessary to better understand the structure-activity relationships of mAbs targeting Pf repetitive elements across life cycle stages, such as the glutamate-rich repeats, and validate these targets as viable for next-generation malaria vaccines seeking the induction of long-lived immunity. The high-throughput target-agnostic approach used here has a strong potential for a further comprehensive exploration of humoral immunity to Pf.

# Materials and methods

**Key resources table**

| Reagent type (species) or resource | Designation | Source or reference | Identifiers | Additional information |
|---|---|---|---|---|
| Strain, strain background (*Plasmodium falciparum*) | NF54 | Radboud University Medical Center; Ponnudurai et al., 1989 | | |
| Strain, strain background (*Anopheles stephensi*) | Nijmegen Sind-Kasur strain | Radboud University Medical Center, Ponnudurai et al., 1989 | | |
| Genetic reagent (*Plasmodium falciparum*) | Pfs48/45 knockout | Radboud University Medical Center; Dijk et al., 2001 | | Pfs48/45 knockout in Pf NF54 background |
| Genetic reagent (*Homo sapiens*) | Fibroblasts expressing CD40L 'L cells' | Laboratory for Immunological Research, Schering-Plough; Garrone et al., 1995 | | |
| Cell line (*Homo sapiens*) | HEK293F | Thermo Fisher Scientific | RRID:CVCL_6642 | |
| Cell line (*Homo sapiens*) | Freestyle 293F | Thermo Fisher Scientific | RRID:CVCL_D615 | |
| Biological sample (*Homo sapiens*) | PBMCs | Radboud University Medical Center; Stone et al., 2018 | | |

*Continued on next page*

*Continued*

| Reagent type (species) or resource | Designation | Source or reference | Identifiers | Additional information |
|---|---|---|---|---|
| Antibody | Anti-human CD3 VioBlue (human monoclonal) | Miltenyi | #130-114-519 | Single B cell sorting (1:50), Recombinant human IgG1 |
| Antibody | Anti-human CD19 PE-Vio 770 (human monoclonal) | Miltenyi | #130-113-647 | Single B cell sorting (1:10), Recombinant human IgG1 |
| Antibody | Anti-human CD20 PE-Vio 770 (human monoclonal) | Miltenyi | #130-111-340 | Single B cell sorting (1:50), Recombinant human IgG1 |
| Antibody | Anti-human CD27 APC (human monoclonal) | Miltenyi | #130-113-636 | Single B cell sorting (1:10), Recombinant human IgG1 |
| Antibody | Anti-human IgM PE (mouse monoclonal) | Miltenyi | #130-093-075 | Single B cell sorting (1:50), Mouse IgG1 |
| Antibody | Anti-human IgD PE (human monoclonal) | Miltenyi | #130-110-643 | Single B cell sorting (1:50), Recombinant human IgG1 |
| Antibody | Anti-human IgA PE (mouse monoclonal) | Miltenyi | #130-113-476 | Single B cell sorting (1:50), Mouse IgG1k |
| Antibody | Anti-human IgG AP (goat polyclonal) | Thermo Fisher Scientific | #A18814 | ELISA (1:2000) |
| Antibody | Alexa Fluor 488 Goat Anti-Mouse IgG (goat polyclonal) | Invitrogen | #A11001 | SIFA (1:200) |
| Antibody | Anti-human IgG-HRP (goat polyclonal) | Pierce | #31412 | Western blot (1:5000), ELISA (1:60,000) |
| Antibody | Anti-Human IgG-TXRD (goat polyclonal) | Southern Biotech | #2040-07 | Microarray, (1:2000) |
| Recombinant DNA reagent | Variable domains of heavy and light chains cloned into gamma1 HC, kappa LC, and lambda LC expression vectors | This paper; *Tiller et al., 2008* | | Inserts are provided in *Supplementary file 2* |
| Recombinant DNA reagent | pCDNA3.4_B1E11K (Fab) | This paper | | Inserts are provided in *Supplementary file 2* |
| Peptide | Pfs230 (P1) | This paper | | Biotin-AHX-AHX-EEVG-EEVG-EEVG-EEVG-GG |
| Peptide | Pfs230 (P2)=RESA P2 (16AA) | This paper | | Biotin-AHX-AHX-EENV-EENV-EENV-EENV-GG |
| Peptide | Pf11.1 (P3) | This paper | | Biotin-AHX-AHX-EELV-EEVIP-EELV-EEFIP-GG |
| Peptide | Pf11.1 (VIP) | This paper | | Biotin-AHX-AHX-EELV-EEVIP-EELV-EE |
| Peptide | Pf11.1 (VVP) | This paper | | Biotin-AHX-AHX-EELV-EEVVP-EELV-EE |
| Peptide | RESA 8AA | This paper | | Biotin-AHX-AHX-EENV-EENV |
| Peptide | RESA 10AA | This paper | | Biotin-AHX-AHX-EENV-EENV-EE |

*Continued on next page*

*Continued*

| Reagent type (species) or resource | Designation | Source or reference | Identifiers | Additional information |
|---|---|---|---|---|
| Peptide | RESA 12AA | This paper | | Biotin-AHX-AHX-EENV-EENV-EENV- |
| Peptide | RESA 14AA | This paper | | Biotin-AHX-AHX-EENV-EENV-EENV-EE |
| Chemical compound, drug | Aqua LIVE/DEAD stain | Thermo Fisher Scientific | #L34957 | |
| Chemical compound, drug | 293Fectin | Thermo Fisher Scientific | #12347500 | Tranfection reagent for mAb expression |
| Chemical compound, drug | Fectopro | Polyplus | #101000014 | Tranfection reagent for Fab expression |
| Chemical compound, drug | ssDNA | Sigma | #D8899-5MG | Polyreactivity testing |
| Chemical compound, drug | Disialoganglioside GD1α | Sigma | #G2392-1MG | Polyreactivity testing |
| Chemical compound, drug | Lipopolysaccharide | Sigma | #L2630-10MG | Polyreactivity testing |
| Chemical compound, drug | Transferrin | Sigma | #T3309-100MG | Polyreactivity testing |
| Chemical compound, drug | Apotransferrin | Sigma | #T1147-100MG | Polyreactivity testing |
| Chemical compound, drug | Hemocyanin | Sigma | #H7017-20MG | Polyreactivity testing |
| Chemical compound, drug | Insulin | Sigma | #I2643-25MG | Polyreactivity testing |
| Chemical compound, drug | Cardiolipin | Sigma | #C0563-10MG | Polyreactivity testing |
| Chemical compound, drug | Histone | Sigma | #H9250-100MG | Polyreactivity testing |
| Chemical compound, drug | Tosyl-activated beads | Invitrogen | #14203 | For immunoprecipitation |
| Chemical compound, drug | SAX biosensors | Sartorius | #18-5117 | For BLI experiments |
| Commercial assay, kit | mRNA TurboCapture kit | QIAGEN | Cat# 72271 | |

## PBMC sampling

Donor A (*Stone et al., 2018*) had lived in Central Africa for approximately 30 years and reported multiple malaria infections during that period. At the time of sampling PBMCs, in 1994, donor A had recently returned to the Netherlands and visited the hospital with a clinical malaria infection. After providing informed consent, PBMCs were collected, but gametocyte prevalence and density were not recorded.

## Preparation of MBCs culture plates

Plates were prepared the day ahead of sorting and incubated at 37°C. Three-hundred-eighty-four-well cell culture plates (Corning #CLS3570-50EA) were prepared with the appropriate memory B cell stimulation media 1 day before cell sorting, to allow the feeder cells sedimentation at the bottom of the wells. Iscove's Modified Dulbecco's Medium (IMDM) (Gibco #12440061) was complemented with 1% penicillin-streptomycin (Thermo Fisher Scientific #10378016) and supplemented with 20% FBS (Gibco #16170-078). A cytokine cocktail was added to stimulate MBCs activation, with IL21 (Preprotech #200-21) at 100 ng/mL and IL2 (Preprotech #200-02) at 51 ng/mL. Fibroblasts expressing CD40L 'L cells' (*Garrone et al., 1995*) were irradiated at 50 Gy and 5000 were added in each well as feeder cells.

## MBCs sorting

Cryopreserved PBMCs were thawed by a brief incubation in a 37°C warm bath and stained with the following Miltenyi REA antibodies before sorting by flow cytometry: anti-CD27 APC (#130-113-636), anti-CD3 VioBlue (#130-114-519), anti-CD19 PE-Vio (#130-113-647), anti-CD20 PE-Vio (#130-111-340), anti-IgM PE (#130-113-476), anti-IgD PE (#130-110-643), and anti-IgA PE (#130-113-476). An Aqua LIVE/DEAD stain was also used (Thermo Fisher Scientific #L34957). Following staining, MBCs were sorted into the 384-well cell culture plates. After an 11-day culture period, supernatants were harvested using a pipetting robot (Eppendorf #5073) and transfered to storage plates (Greiner #788860-906). MBCs were lysed and their mRNA purified using the mRNA TurboCapture kit for 384 wells (QIAGEN #72271). Lysates were stored at –80°C. Lysates from selected wells were further transferred into 96-well RT-PCR plates (Bio-Rad #HSP9641) to perform RT-PCR.

## Gamete/gametocyte extract ELISA

Gamete or gametocyte lysate were prepared as described (*Fabra-García et al., 2023*). 384-well plates (Thermo Fisher Scientific #460372) were coated with 7500 lysed gametes/gametocytes per well. Plates were incubated at 4°C overnight, then wells were washed three times with PBS-Tween 0.05%, prior to 1 hr blocking (with a 1% BSA-1% PBS-Tween 0.05% solution). Cell culture supernatants (diluted twofold in blocking solution) were dispensed into the wells for a 1 hr incubation step. Following washing, anti-human IgG AP antibody (Thermo Fisher Scientific #A18814) diluted at 1/2000 was added and incubated for 1 hr. Plates were then washed and 15 µl of CDP-substrate (Thermo Fisher Scientific #T2146) was added. The reaction was measured using Biotek Synergy 2 reader. Positivity threshold was determined as the average background OD + (3× SD (background OD)). 0.3 µg/mL TB31F (anti-Pfs48/45 mAb) and 1.0 µg/mL 2544 (anti-*Pf*s25 mAb) were used as positive controls, while 30.0 µg/mL 399 (anti-CSP mAb) was used as negative control.

## mAb isolation and production

Nested multiplexed PCRs were performed on single MBCs from selected wells following the protocol outlined by *Tiller et al., 2008*. PCR products were sent for sequencing (Genewiz) and sequences analyzed for Ig gene features using the IMGT (ImMunoGeneTics) database (*Lefranc et al., 2015*). Ig gene family-specific primers were used for cloning, as described by *Tiller et al., 2008*. Purified PCR products were cloned into vectors encoding for either IgG1 lambda, kappa, or heavy constant regions. For transient mAb expression and secretion, HEK293F cells were co-transfected with plasmids coding the antibody heavy chain and the corresponding light chain using 293Fectin (#12347500). A protein A-Sepharose column (Sigma #ge17-1279-03) was used for mAb purification. Elution of mAbs was conducted with 4.5 mL of glycine 0.1 M (pH 2.5) and 500 µl of Tris 1 M (pH 9). The purified mAbs were subsequently subjected to buffer exchange and concentration with AmiconUltra (Merck #36100101).

## Fab production

The DNA sequences of VK and VH of the B1E11K Fab domain were cloned upstream of human Igκ and Igγ1-CH1 domains respectively and inserted into a custom pcDNA3.4 expression vector. The plasmids were co-transfected into FreeStyle 293 F cells that were cultured in FreeStyle 293 Expression Media (Gibco #12338018) using Fectopro (Polyplus, 101000014). The recombinant Fab was purified via KappaSelect affinity chromatography (Cytiva #17545812) and cation exchange chromatography (MonoS, Cytiva 17516801).

## Gamete SIFA

Gamete SIFA was performed with *Pf* NF54 wild-type and Pfs48/45 knockout (*Dijk et al., 2001*) strains. The Pfs48/45 knockout lacks both surface-bound Pfs48/45 and Pfs230 (*Stone et al., 2018*; *Eksi et al., 2006*; *Kumar, 1987*). Wild-type or Pfs48/45 knockout gametes were obtained following gametocyte activation in FBS for 1 hr at room temperature. Gametes were washed with PBS and incubated with mAbs diluted in PBS containing 0.5% PBS and 0.05% NaN₃ (SIFA buffer) for 1 hr at 4°C in sterile V bottom plates (VWR#736-0223). After incubation, wells were washed three times with SIFA buffer and secondary antibody Alexa Fluor 488 Goat Anti-Mouse IgG (H+L) (Invitrogen#A11001) diluted 1/200 added for a 1 hr incubation step on ice. Following a washing step, gametes were suspended in 4%

paraformaldehyde and transferred into 384-well clear bottom black plates. Four images per well were taken using the ImageXpress Pico Cell Imaging System (Molecular Devices).

## Western blot

For western blots with gametocyte extract, *Pf* NF54 gametocyte extract was prepared as described above. The extract was mixed with NuPAGE LDS sample buffer (Thermo Fisher Scientific # NP0008) and heated for 15 min at 56°C. The equivalent of 1 million lysed gametocytes was loaded per lane. A NuPAGE 4–12% Bis-Tris 2D-well gel (Thermo Fisher Scientific #NP0326BOX) was used for proteins separation. Using the Trans-Blot Turbo system (Bio-Rad #1704150) samples were then transferred to a 0.22 µm nitrocellulose membrane (Bio-Rad #1620150). The blots were cut into strips, blocked with 5% skimmed milk in PBS and incubated with 5 µg/mL of the mAb to be tested. Strips were incubated with the secondary anti-human IgG-HRP antibody (Pierce #31412), diluted 1/5000 in PBS-T. Clarity Western ECL substrate (Bio-Rad #1705060) was used for development and strips were imaged with the ImageQuant LAS4000 equipment (GE HealthCare).

For western blot with recombinant proteins, we used Pfs230CMB (amino acids 444–730) expressed in a plant-based transient expression system (*Farrance et al., 2011*), and RESA3 (amino acids 570–1090), RESA (amino acids 66–585), LSA3 (amino acids 805–1558), and Pf11.1 (amino acids 3657–3734) that were expressed wheat germ cell free extract (https://repository.ubn.ru.nl/bitstream/handle/2066/289602/289602.pdf?sequence=1). All antigens contained a C-terminal His-tag. RESA and LSA3 also had an N-terminal GST-tag. The equivalent of 150 ng of protein was loaded per lane. An SDS 4–20% gel (Bio-Rad # 4561094) was used for protein separation under non-reducing conditions. Further steps were performed following the protocol described above.

## Recombinant Pfs48/45 and Pfs230 ELISA

ELISAs with full-length Pfs48/45 and fragments thereof, and Pfs230CMB were performed as previously described (*Fabra-García et al., 2023*; *Farrance et al., 2011*). In short, Nunc MaxiSorp 96-wells plates (Thermo Fisher) were coated with 0.5 µg/mL recombinant Pfs48/45 or Pfs230CMB proteins, blocked with 5% skimmed milk in PBS+0.1% Tween-20 and washed. Plates were then incubated with 10 µg/mL or 30 µg/mL mAb in 1% skimmed milk in PBS. After washing, plates were incubated with 1:60,000 Goat Anti-Human IgG/HRP-conjugated antibody (Pierce, #31412) in 1% skimmed milk in PBS+0.1% Tween-20. After washing, plates were developed with 3,3',5,5'-tetramethylbenzidine and the reaction was stopped with $H_2SO_4$. Absorbance was measured at 450 nm. mAbs were considered positive when the absorbance was higher than the mean absorbance plus three standard deviations of seven negative mAbs.

## MAb polyreactivity testing in ELISA

The coating antigens were diluted to 1 µg/mL. Antigens used were: ssDNA (Sigma #D8899-5MG), disialoganglioside GD1α (Sigma #G2392-1MG), LPS (Sigma #L2630-10MG), transferrin (Sigma #T3309-100MG), apotransferrin (Sigma #T1147-100MG), hemocyanin (Sigma #H7017-20MG), insulin (Sigma #I2643-25MG), cardiolipin (Sigma #C0563-10MG), albumin and histone (Sigma #H9250-100MG). Secondary antibody used was a phosphatase-coupled goat anti-human IgG (Jackson ImmunoResearch #109 056 098). Optical densities were read at 405 nm, 1 hr after the addition of pNPP.

## SMFA

SMFA experiments were performed using *Pf* NF54 wild-type gametocytes with oocyst count readout, following a protocol set up by *Ponnudurai et al., 1989*. Briefly, blood meals containing cultured gametocytes mixed with antibodies were fed to *A. stephensi* mosquitoes (Nijmegen colony). For each condition, 20 fully fed mosquitoes were analyzed. Reported antibody concentrations are concentrations in the total blood meal volume. mAbs that showed >80% TRA, i.e., reduction in oocysts compared to a negative control, were tested in a second independent SMFA experiment. TRA from one or two independent SMFA experiments was calculated using a negative binomial regression model as previously described (*de Jong et al., 2022b*). SMFA data analyses were done in R (version 4.1.2).

## Microarray

Microarray design and protocol have been extensively detailed in *Stone et al., 2018*, and *de Jong et al., 2022a*. Briefly, the selection of proteins to be printed on the array was made on the basis of a

systematic analyses of proteomic data by *Meerstein-Kessel et al., 2018*. In total, 943 protein targets representing 528 unique gene IDs were expressed for the array using an in vitro transcription translation system; these were printed onto nitrocellulose-coated slides at the University of California, Irvine, as described previously (*de Jong et al., 2022a*). Microarray slides were rehydrated in blocking buffer (GVS #10485356) while B1E11K was diluted 1:100 in a 20% *Escherichia coli* lysate/blocking buffer solution and incubated for 30 min. Blocking buffer was discarded and diluted B1E11K added to the array slides for incubation overnight at 4°C with continual rocking. Following three washes with TBS-Tween-20 0.05%, slides were probed with a fluorophore-conjugated secondary antibody (Southern Biotech, Goat Anti-Human IgG-TXRD) at a concentration of 0.5 µg/mL (1:2000) in 2% *E. coli* lysate/blocking buffer solution. After three washes, slides were removed from their casettes and rinsed in ddH$_2$O air dried in a centrifuge and scanned using a GenePix 4300A High-resolution Microarray Scanner (Molecular Devices). Data treatment and analysis were performed using R (*Staples, 2023*). Correction for local array target spot background was done using the 'backgroundCorrect' function of the limma package (*Ritchie et al., 2015*). Background corrected values were log2 transformed and normalized to systematic effects by subtraction of the median signal intensity of the negative IVTT controls (internally within four subarrays per sample). The final normalized data are a log2 MFI ratio of target to control reactivity: a value of 0 represents equality with the vehicle control, and a value of 1 indicates a signal twice as high.

## Immunoprecipitation

Briefly, we used tosyl-activated beads (Invitrogen #14203) to covalently link B1E11K and incubated these beads with a gametocyte lysate to enable antigen capture. Immunoprecipitated antigens were eluted, and the elution fraction was run on an SDS-PAGE gel and silver-stained. A negative control immunoprecipitation experiment was performed using an anti-HIV gp120 mAb. As shown in sup *Figure 3—figure supplement 1B*, two bands with a molecular weight greater than 250 kDa, and a third one with a molecular weight around 55 kDa were specifically detected in the B1E11K immunoprecipitate, in comparison to the control antibody. The three bands were cut and sent for mass spectrometry analysis. Data were analyzed by querying the entire proteome of *Pf* (NF54 isolate) in the UniProt database.

## Peptide synthesis

Peptides were produced by P. Verdié team, IBMM – SynBio3, Montpellier, France. Lyophilized peptides were solubilized in PBS.

## ELISA with biotinylated peptides

ELISA protocol was similar to the protocol described above. Briefly, 96-well ELISA plates were coated overnight at 4°C with 0.5 µg/mL of streptavidin (Thermo Fisher Scientific # 434301). Plates were washed and then blocked for 1 hr. Following wash, peptides diluted at 0.5 µg/mL were added to the plates for a 1 hr incubation. The plates were washed and serially diluted mAbs were added. mAb fixation was detected using phosphatase-coupled goat anti-human IgG (Jackson ImmunoResearch #109 056 098) and para-nitrophenylphosphate (Interchim #UP 664791). The enzymatic reaction was measured at 405 nm using a TECAN Spark 10M plate reader. Half maximal effective concentration (EC50) was calculated from raw data (OD) after normalization using GraphPad Prism (version 9) 'log agonist versus normalized response' function.

## BLI assay

All BLI experiments were performed using an Octet Red96e Instrument (PallForteBio), at 25°C and under 1000 rpm agitation. SAX biosensors (Sartorius #18-5117) were pre-wetted in BLI Buffer (PBS [pH 7.4]+0.01% [wt/vol] BSA +0.002% [vol/vo] Tween-20) for 10 min. Biotinylated peptides were loaded onto the biosensors until the top concentration of B1E11K Fab utilized in kinetic assays (2500 nM for RESA 10AA peptide and 625 nM for RESA P2 peptide) yielded a response value of ~1.2 nm. An association step was conducted by dipping the sensors into a titration series of ½ serially diluted B1E11K Fab for 30 s. The dissociation step was conducted by dipping the biosensors into BLI Buffer for 1200s. Background subtractions were done using measurements where experiments were performed with

biosensors treated in the same conditions but replacing Fab solution with BLI Buffer. Kinetic data were processed using the manufacturer's software (Data Analysis HT v11.1).

## Isothermal titration calorimetry

An Auto-ITC200 (Malvern) was used to conduct calorimetric experiments. The RESA P2 peptide, RESA 10AA peptide, and recombinant B1E11K Fab were buffer-exchanged into Tris-buffered saline (20 mM Tris pH 7.0, and 150 mM NaCl). The B1E11K Fab was concentrated at 90–110 µM for experiments utilizing the RESA P2 peptide and 60–70 µM for those utilizing the RESA 10AA peptide. The RESA P2 peptide and RESA 10AA peptide were concentrated at 5–6 µM and 7–10 µM respectively. Fab (syringe) was titrated into the cell (peptide) at 25°C using a protocol involving 19 injections each at a volume of 2.0 µl. The curves were fitted to a 2:1 or 1:1 binding model using the MicroCal ITC Origin 7.0 Analysis Software.

## SEC-MALS

SEC-MALS experiments were performed at 4°C using a Superdex 200 Increase 10/300 GL (Cytiva #GE17-5175-01) column. RESA P2 peptide was incubated with B1E11K Fab at a 1:6 molar ratio for 30 min prior to loading onto the Superdex 200 column. The column was set up onto an Agilent Technologies 1260 Infinity II HPLC coupled with a MiniDawn Treos MALS detector (Wyatt), Quasielastic light scattering (QELS) detector (Wyatt), and Optilab T-reX refractive index (RI) detector (Wyatt). Data processing was performed using the ASTRA software (Wyatt).

## nsEM and image processing

Fractions of the first peak of the SEC-MALS experiments containing the 2:1 B1E11K:RESA P2 complex were used to make nsEM grids. 50 µg/mL of the complex was deposited onto homemade carbon film-coated grids (previously glow-discharged in air for 15 s) and stained with 2% uranyl formate. Data was collected onto a Hitachi HT7800 microscope paired with an EMSIS Xarosa 20 Megapixel CMOS camera. Micrographs were taken with the microscope operating at 120 kV at ×80,000 magnification with a pixel size of 1.83 Å/px. Image processing, particle picking, extractions, 2D classifications, and 3D reconstructions were done in cryoSPARC v2 (*Punjani et al., 2017*).

## X-ray crystallography and structural determination

The RESA P2 peptide was incubated with B1E11K Fab at a 1:5 molar ratio for 30 min prior to loading onto a Superdex 200 column Increase 10/300 GL column. Fractions containing the complex were pooled and concentrated at 8.6 mg/mL. A seed stock prepared from a previous crystallization trial was used for seeding. The stock was prepared from condition G9 of a JCSG Top96 screen (0.2 M (NH$_4$)2SO$_4$ 25% [wt/vol] PEG4000, and 0.1 M sodium acetate [pH 4.6]). The complex, reservoir solution, and seed stock were mixed at a 3:4:1 volumetric ratio into an optimization tray derived from condition G9 of the JCSG Top96 screen. Crystals grew within 6 hr in a reservoir condition consisting of (0.1 M NH$_4$)2SO$_4$ 25% (wt/vol) PEG4000, and 0.1 M sodium acetate (pH 5.2). Crystals were cryo-protected with 15% ethylene glycol (vol/vol) before being flash-frozen in liquid nitrogen. Data collection was performed at the 23-ID-B beamline at the Argonne National Laboratory Advanced Photon Source. Datasets were initially processed using autoproc (*Vonrhein et al., 2011*) and further optimized using xdsgui (*Kabsch, 2010*). Molecular replacement was performed using PhaserMR *McCoy et al., 2007* followed by multiple rounds of refinement using phenix.refine (*Liebschner et al., 2019*) and Coot (*Emsley et al., 2010*). Inter- and intra-molecular contacts were determined using PISA (*Krissinel and Henrick, 2007*) and manual inspection. Structural figures were generated using UCSF ChimeraX (*Goddard et al., 2018*; *Pettersen et al., 2021*).

## Material availability statement

Materials generated in this study will be made available upon reasonable request.

## Acknowledgements

We thank A Guarino for her support to antibody production, N Thielens for her input during the course of this work, Y Couté for the proteomics analysis, J-B Reiser for the preliminary BLI experiments, L Chaperot for providing CD40L expressing fibroblasts, E Thai and D Ivanochko for their

contributions to X-ray data collection and structure determination, K Teelen for assistance with ELISAs, M van de Vegte-Bolmer and R Stoter for parasite culture, GJ van Gemert, L Pelser, A Pouwelsen, J Kuhnen, J Klaassen, and S Mulder for mosquito rearing and dissection, and K Koolen for support with ELISA and gamete binding experiments. Furthermore, we would like to thank Jessica Chichester (Fraunhofer) for providing Pfs230CMB and P Felgner (UCI) for production of protein microarrays. JPJ was supported by the CIFAR Azrieli Global Scholar program, the Ontario Early Researcher Award program and the Canada Research Chair program. RY was supported by a Canada Graduate Scholarship – Master's (CGS-M). The proteomic experiments were partially supported by ANR grant ProFI (Proteomics French Infrastructure, ANR-10-INBS-08) and GRAL, a program from the Chemistry Biology Health (CBH) Graduate School of University Grenoble Alpes (ANR-17-EURE-0003). This work used the platforms of the Grenoble Instruct-ERIC center (ISBG; UAR 3518 CNRS-CEA-UGA-EMBL) within the Grenoble Partnership for Structural Biology (PSB), supported by FRISBI (ANR-10-INBS-0005–02) and GRAL, financed within the University Grenoble Alpes graduate school (Ecoles Universitaires de Recherche) CBH-EUR-GS (ANR-17-EURE-0003). Molecular graphics were generated using UCSF ChimeraX, developed by the Resource for Biocomputing, Visualization, and Informatics (University of California, San Francisco) with support from the National Institutes of Health (R01-GM129325) and the Office of Cyber Infrastructure and Computational Biology, National Institute of Allergy and Infectious Diseases.The BLI, ITC, and SEC-MALS instruments were accessed at the Structural and Biophysical Core Facility, The Hospital for Sick Children, and EM data was collected at the Nanoscale Biomedical Imaging Facility. The Hospital for Sick Children, supported by the Canada Foundation for Innovation and Ontario Research Fund X-ray diffraction experiments, were in part performed using beamlines 23-ID-B at GM/CA@APS, which has been funded by the National Cancer Institute (ACB-12002) and the National Institute of General Medical Sciences (AGM-12006, P30GM138396). This research used resources of the Advanced Photon Source, a U.S. Department of Energy (DOE) Office of Science User Facility operated for the DOE Office of Science by Argonne National Laboratory under Contract No. DE-AC02-06CH11357. The Eiger 16M detector at GM/CA-XSD was funded by NIH grant S10 OD012289 and X-ray diffraction experiments were also performed using beamline AMX-17-ID-1 at the National Synchrotron Light Source II, a U.S. Department of Energy (DOE) Office of Science User Facility operated for the DOE Office of Science by Brookhaven National Laboratory under Contract No. DE-SC0012704. The Center for BioMolecular Structure (CBMS) is primarily supported by the National Institutes of Health, National Institute of General Medical Sciences (NIGMS) through a Center Core P30 Grant (P30GM133893), and by the DOE Office of Biological and Environmental Research (KP1607011). X-ray diffraction experiments were also performed using beamline CMCF-ID at the Canadian Light Source, a national research facility of the University of Saskatchewan, which is supported by the Canada Foundation for Innovation (CFI), the Natural Sciences and Engineering Research Council (NSERC), the National Research Council (NRC), the Canadian Institutes of Health Research (CIHR), the Government of Saskatchewan, and the University of Saskatchewan.

## Additional information

### Funding

| Funder | Grant reference number | Author |
| --- | --- | --- |
| Bill and Melinda Gates Foundation | OPP1108403 | Axelle Amen<br>Amanda Fabra-García<br>Judith Bolscher<br>Isabelle Bally<br>Sebastián Dergan-Dylon<br>Roos M de Jong<br>Marloes de Bruijni<br>C Richter King<br>Randall S MacGill<br>Robert W Sauerwein<br>Pascal Poignard<br>Matthijs M Jore |

| Funder | Grant reference number | Author |
| --- | --- | --- |
| Canadian Institutes of Health Research | 428410 | Jean-Philippe Julien<br>Randy Yoo |
| Wellcome Trust | 218676/Z/19/Z | William JR Stone |
| Nederlandse Organisatie voor Wetenschappelijk Onderzoek | VIDI fellowship number 192.061 | Matthijs M Jore |

The funders had no role in study design, data collection and interpretation, or the decision to submit the work for publication. For the purpose of Open Access, the authors have applied a CC BY public copyright license to any Author Accepted Manuscript version arising from this submission.

## Author contributions

Axelle Amen, Randy Yoo, Conceptualization, Formal analysis, Investigation, Writing – original draft; Amanda Fabra-García, Judith Bolscher, Conceptualization, Formal analysis, Investigation, Writing – review and editing; William JR Stone, Isabelle Bally, Sebastián Dergan-Dylon, Iga Kucharska, Roos M de Jong, Marloes de Bruijni, Formal analysis, Investigation, Writing – review and editing; Teun Bousema, C Richter King, Randall S MacGill, Conceptualization, Funding acquisition, Writing – review and editing; Robert W Sauerwein, Conceptualization, Formal analysis, Supervision, Funding acquisition, Writing – review and editing; Jean-Philippe Julien, Pascal Poignard, Matthijs M Jore, Conceptualization, Formal analysis, Supervision, Funding acquisition, Writing – original draft

## Author ORCIDs

Axelle Amen ⓘ https://orcid.org/0000-0002-0449-4445
Randy Yoo ⓘ http://orcid.org/0000-0002-6952-9039
Amanda Fabra-García ⓘ http://orcid.org/0000-0001-6663-213X
Judith Bolscher ⓘ http://orcid.org/0000-0002-1898-6096
William JR Stone ⓘ http://orcid.org/0000-0002-6647-0166
Teun Bousema ⓘ https://orcid.org/0000-0003-2666-094X
Randall S MacGill ⓘ http://orcid.org/0000-0002-4566-1481
Jean-Philippe Julien ⓘ https://orcid.org/0000-0001-7602-3995
Pascal Poignard ⓘ https://orcid.org/0000-0002-0021-7192
Matthijs M Jore ⓘ https://orcid.org/0000-0002-0686-370X

## Ethics

At the time of sampling PBMCs, in 1994, Donor A had recently returned to the Netherlands and visited the hospital with a clinical malaria infection. After providing informed consent, PBMCs were collected.

Reviewer #2 (Public Review): https://doi.org/10.7554/eLife.97865.3.sa1
Reviewer #3 (Public Review): https://doi.org/10.7554/eLife.97865.3.sa2
Author response https://doi.org/10.7554/eLife.97865.3.sa3

# Additional files

## Supplementary files

Supplementary file 1. Genetic characteristics of isolated antibodies.

Supplementary file 2. Sequences of isolated antibodies.

Supplementary file 3. PDB validation report.

Supplementary file 4. Interactions between B1E11K Fabs and RESA P2 (16AA) peptide.

Supplementary file 5. Homotypic interactions between B1E11K Fab A and Fab B.

MDAR checklist

## Data availability

All data generated or analysed during this study are included in the manuscript and supporting files. Microarray data, raw SMFA data and Antibody sequences are provided as source data and supplementary files. Diffraction data have been deposited in PDB under the accession code 8US8.

The following dataset was generated:

| Author(s) | Year | Dataset title | Dataset URL | Database and Identifier |
|-----------|------|---------------|-------------|-------------------------|
| Yoo R, Julien JP | 2024 | Crystal structure of B1E11K malarial antibody in complex with RESA repeat peptide | https://www.rcsb.org/structure/8US8 | RCSB Protein Data Bank, 8US8 |

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
