## [Editor Report · eLife assessment]

This study reports **important** results and new insights into humoral immune responses to *Plasmodium falciparum* sexual stage proteins. The experiments are based on the use of target-agnostic memory B cell sorting and screening approaches as well as several state-of-the-art technologies. The authors present **compelling** evidence that one antibody, B1E11K, is cross-reactive with multiple proteins containing glutamate-rich repeats through homotypic interactions, a process similar to what has been observed for *Plasmodium* circumsporozoite protein repeat-directed antibodies.

---

## [Referee Report · Reviewer #2 (Public Review)]

This manuscript by Amen, Yoo and Fabra-Garcia et al describes a human monoclonal antibody B1E11K, targeting EENV repeats which are present in parasite antigens such as Pfs230, RESAs and Pf11.1. The authors isolated B1E11K using an initial target agnostic approach for antibodies that would bind gamete/gametocyte lysate which they made 14 mAbs. Following a suite of highly appropriate characterization methods from Western blotting of recombinant proteins to native parasite material, use of knockout lines to validate specificity, ITC, peptide mapping, SEC-MALS, negative stain EM and crystallography, the authors have built a compelling case that B1E11K does indeed bind EENV repeats. In addition, using X-ray crystallography they show that two B1E11K Fabs bind to a 16 aa RESA repeat in a head-to-head conformation using homotypic interactions and provide a separate example from CSP, of affinity-matured homotypic interactions.

The authors have addressed most of our previous comments in their revised manuscript.

One of the main conclusions in the paper is the binding of B1E11K to RESAs which are blood stage antigens that are exported to the infected parasite surface. In the future, it would be interesting to understand if B1E11K mAb binds to the red cell surface of infected blood stage parasites to understand its cellular localization in those stages.

Materials and Methods:

PBMC sampling: While the authors have provided clarification that they obtained informed consent from the PBMC donor, they have not added the ethics approval codes in this section.

---

## [Referee Report · Reviewer #3 (Public Review)]

The manuscript from Amen et al reports the isolation and characterization of human antibodies that recognize proteins expressed at different sexual stages of *Plasmodium falciparum*. The isolation approach was antigen agnostic and based on the sorting, activation, and screening of memory B cells from a donor whose serum displays high transmission-reducing activity. From this effort, 14 antibodies were produced and further characterized. The antibodies displayed a range of transmission-reducing activities and recognized different Pf sexual stage proteins. However, none of these antibodies had substantially higher TRA than previously described antibodies.

The authors then performed further characterization of antibody B1E11K, which was unique in that it recognized multiple proteins expressed during sexual and asexual stages. Using protein microarrays, B1E11K was shown to recognize glutamate-rich repeats, following an EE-XX-EE pattern. An impressive set of biophysical experiments were performed to extensively characterize the interactions of B1E11K with various repeat motifs and lengths. Ultimately, the authors succeeded in determining a 2.6 A resolution crystal structure of B1E11K bound to a 16AA repeat-containing peptide. Excitingly, the structure revealed that two Fabs bound simultaneously to the peptide and made homotypic antibody-antibody contacts. This had only previously been observed before with antibodies directed against CSP repeats.

Overall I found the manuscript to be very well written. Strengths of the manuscript include the target-agnostic screening approach and the thorough characterization of antibodies. The demonstration that B1E11K is cross-reactive to multiple proteins containing glutamate-rich repeats, and that the antibody recognizes the repeats via homotypic interactions, similar to what has been observed for CSP repeat-directed antibodies, should be of interest to many in the field.

---

## [Author Response]

The following is the authors’ response to the original reviews.

**Public Reviews:**

**Reviewer #1 (Public Review):**
Summary:In this paper, the authors used target agnostic MBC sorting and activation methods to identify B cells and antibodies against sexual stages of *Plasmodium falciparum*. While they isolated some Mabs against PFs48/45 and PFs230, two well-known candidates for "transmission blocking" vaccines, these antibodies' efficacies, as measured by TRA, did not perform as well as other known antibodies. They also isolated one cross-reactive mAb to proteins containing glutamic acid-rich repetitive elements, that express at different stages of the parasite life cycle. They then determined the structure of the Fab with the highest protein binder they could determine through protein microarray, RESA, and observed homotypic interactions.Strengths:- Target agnostic B cell isolation (although not a novel methodology).- New cross-reactive antibody with some "efficacy" (TRA) and mechanism (homotypic interactions) as demonstrated by structural data and other biophysical data.Weaknesses:The paper lacks clarity at times and could benefit from more transparency (showing all the data) and explanations.

We have added the oocyst count data from the SMFA experiments as Supplementary Table 2, and ELISA binding curves underlying Figure 4B as Supplementary Figure 5.

In particular:- define SIFA- define TRAbs

We have carefully gone through the manuscript and have introduced abbreviations at first use, removed unnecessary abbreviations and removed unnecessary jargon to increase readability.

- it is not possible to read the Figure 6B and C panels.

We regret that the labels in Supplementary Figures 6 and 7 were of poor quality and have now included higher resolution images to solve this issue.

**Reviewer #2 (Public Review):**
This manuscript by Amen, Yoo, Fabra-Garcia et al describes a human monoclonal antibody B1E11K, targeting EENV repeats which are present in parasite antigens such as Pfs230, RESAs, and 11.1. The authors isolated B1E11K using an initial target agnostic approach for antibodies that would bind gamete/gametocyte lysate which they made 14 mAbs. Following a suite of highly appropriate characterization methods from Western blotting of recombinant proteins to native parasite material, use of knockout lines to validate specificity, ITC, peptide mapping, SEC-MALS, negative stain EM, and crystallography, the authors have built a compelling case that B1E11K does indeed bind EENV repeats. In addition, using X-ray crystallography they show that two B1E11K Fabs bind to a 16 aa RESA repeat in a head-to-head conformation using homotypic interactions and provide a separate example from CSP, of affinity-matured homotypic interactions.There are some minor comments and considerations identified by this reviewer, These include that one of the main conclusions in the paper is the binding of B1E11K to RESAs which are blood stage antigens that are exported to the infected parasite surface. It would have been interesting if immunofluorescence assays with B1E11K mAb were performed with blood-stage parasites to understand its cellular localization in those stages.

In the current manuscript, we provide multiple lines of evidence that B1E11K binds (with high affinity) to repeats that are present in RESAs, i.e. through micro-array studies, in vitro binding experiments such as Western blot, ELISA and BLI, and through X-ray crystallography studies on B1E11k – repeat peptide complexes. Taken together, we think we provide compelling evidence that B1E11k binds to repeats present in RESA proteins. We do agree that studies on the function of this mAb against other stages of the parasite could be of interest, but as our manuscript focuses on the sexual stage of the parasite, we feel that this is beyond scope of the current work. However, this line of inquiry will be strongly considered in follow up studies.

**Reviewer #3 (Public Review):**
The manuscript from Amen et al reports the isolation and characterization of human antibodies that recognize proteins expressed at different sexual stages of *Plasmodium falciparum*. The isolation approach was antigen agnostic and based on the sorting, activation, and screening of memory B cells from a donor whose serum displays high transmission-reducing activity. From this effort, 14 antibodies were produced and further characterized. The antibodies displayed a range of transmission-reducing activities and recognized different Pf sexual stage proteins. However, none of these antibodies had substantially lower TRA than previously described antibodies.The authors then performed further characterization of antibody B1E11K, which was unique in that it recognized multiple proteins expressed during sexual and asexual stages. Using protein microarrays, B1E11K was shown to recognize glutamate-rich repeats, following an EE-XX-EE pattern. An impressive set of biophysical experiments was performed to extensively characterize the interactions of B1E11K with various repeat motifs and lengths. Ultimately, the authors succeeded in determining a 2.6 A resolution crystal structure of B1E11K bound to a 16AA repeat-containing peptide. Excitingly, the structure revealed that two Fabs bound simultaneously to the peptide and made homotypic antibody-antibody contacts. This had only previously been observed with antibodies directed against CSP repeats.Overall I found the manuscript to be very well written, although there are some sections that are heavy on field-specific jargon and abbreviations that make reading unnecessarily difficult. For instance, 'SIFA' is never defined.

We have carefully gone through the manuscript and have introduced abbreviations at first use, removed unnecessary abbreviations and removed unnecessary jargon to increase readability.

Strengths of the manuscript include the target-agnostic screening approach and the thorough characterization of antibodies. The demonstration that B1E11K is cross-reactive to multiple proteins containing glutamate-rich repeats, and that the antibody recognizes the repeats via homotypic interactions, similar to what has been observed for CSP repeat-directed antibodies, should be of interest to many in the field.
**Recommendations for the authors:**

**Reviewer #1 (Recommendations For The Authors):**
Figure 1 - why only gametes ELISA and Spz or others?

The volumes of the single B cell supernatants were too small to screen against multiple antigens/parasite stages. As we aimed to isolate antibodies against the sexual stages of the parasite, our assay focused on this stage and supernatants were not tested against other stages. Furthermore, we screened for reactivity against gametes as TRA mAbs likely target gametes rather than other forms of sexual stage parasites.

Figure 2 A(a) Wild type (WT) and Pfs48/45 knock-out (KO) gametes.(b) I am a bit confused about what GMT is vs Pfs48/45

We have changed the column titles in Figure 2A to 'wild-type gametes' and 'Pfs48/45 knockout gametes' to improve clarity.

(c) Binding is high % why is it red?

We chose to present the results in a heatmap format with a graded color scale, from strong binders in red to weak binders in green. It has now been clarified in the legend of the figure.

Please state acronyms clearlyTRA - transmission reducing activitySMFA - standard membrane feeding assay

We have added the full terms to clarify the acronyms.

1123- VRC01 (not O1)

We have corrected this.

Figure 2 C bottom panels, clarify which ones are TRAbs (Assuming the Mabs with over 80% TRA at 500 ug/ml) (right gel) and the ones that are not (left gel)?

In the Western blot in Figure 2c, we have marked the antibodies with >80% TRA with an asterisk.

Furthermore, we have replaced ‘TRAbs’ by ‘mAbs with >80% TRA at 500 µg/mL’ in the figure legend.

ITC show the same affinity of the Fab to the 2 peptides but not the ELISA, not the BLI/SPR would be more appropriate. Any potential explanation?

The way binding affinity is determined across various techniques can result in slight differences in determined values. For instance, ELISAs utilize long incubation times with extensive washing steps and involve a spectroscopic signal, isothermal titration calorimetry (ITC) uses calorimetric signal at different concentration equilibriums to extract a KD, and BLI determines kinetic parameters for KD determination. Discrepancies in binding affinities between orthologous techniques have indeed been observed previously in the context of peptide-antibody binding (e.g. PMID: 34788599).

Despite this, regardless of technique, the relative relationships in all three sets of data is the same - higher binding affinity is observed to the longer P2 peptide. This is the main takeaway of the section. As the reviewer suggests, BLI is likely the most appropriate readout here and is the only value explicitly mentioned in the main text. We primarily use ITC to support our proposed binding stoichiometry which is important to substantiate the SEC-MALS and nsEM data in Figure 4H-I. We added the following sentences to help reinforce these points: 'The determined binding affinity from our ITC experiments (Table 1) differed from our BLI experiments (Fig. 4D and 4E), which can occur when measuring antibody-peptide interactions. Regardless, our data across techniques all trend toward the same finding in which a stronger binding affinity is observed toward the longer RESA P2 (16AA) peptide.'

Figure 5C - would be helpful to have the peptide sequence above referring to what is E1, E2 etc...

We added two panels (Figure 5C-D) showcasing the binding interface that shows the peptide numbering in the context of the overall complex. We hope that this will help better orient the reader.

Figure S4 - maybe highlight in different colors the EENVV, EEIEE, Etc, etc

Repeats found in the sequence of the various proteins in Figure S4 have now been highlighted with different colors.

Line 163 - why 14 mabs if 11 wells? Isn't it 1 B cell per well? The authors should explain right away that some wells have more than 1 B cell and some have 1 HC, 1LC, and 1 KC.

We agree that this was somewhat confusing and have modified the text which now reads: 'We obtained and cloned heavy and light chain sequences for 11 out of 84 wells. For three wells we obtained a kappa light chain sequence and for five wells a lambda light chain sequence. For three wells we obtained both a lambda and kappa light chain sequence suggesting that either both chains were present in a single B cell or that two B cells were present in the well. For all 14 wells we retrieved a single heavy chain sequence. Following amplification and cloning, 14 mAbs, from 11 wells, were expressed as full human IgG1s (Table S1) (Dataset S1).'

Line 166-167 - were they multiple HC (different ones) as well when Lambda and kappa were present?This is not clear at first.

We clarified this point in the text, see also comment above.

Line 177 - expressed Pfs48/45 and Pfs230, is it lacking both or just Pfs48/45 (as stated on line 172)?

Pfs48/45 binds to the gamete surface via a GPI anchor, while Pfs230 is retained to the surface through binding to Pfs48/45. Hence, the Pfs48/45 knockout parasite will therefore also lack surfacebound Pfs230. We have added a sentence to the Results clarifying this: 'The mAbs were also tested for binding to Pfs48/45 knock-out female gametes, which lack surface-bound Pfs48/45 and Pfs230'.

Show the ELISA data used to calculate EC50 in Figure 3.

ELISA binding curves are now shown as Figure S5.

Line 313-315 - what if you reverse, capture the Fab (peptide too small even if biotinylated?)

As anticipated by the Reviewer, immobilizing the Fab and dipping into peptide did not yield appreciable signal for kinetic analysis and thus the experiment from this setup is not reported.

Line 341 - add crystal structure

This has now been added.

There is a bit too much speculation in the discussion. For e.g. "The B1C5L and B1C5K mAbs were shown to recognize Domain 2 of Pfs48/45 and exhibited moderate potency, as previously described for Abs with such specificity (27). These 2 mAbs were isolated from the same well and shared the same heavy chain; their three similar characteristics thus suggest that their binding is primarily mediated by the heavy chain". Actual data will reinforce this statement.

As B1C5L and B1C5K recognized domain 2 of Pfs48/45 with similar affinity, this strongly suggests that binding is mediated though the heavy chain. Structural analysis could confirm this statement, but this is out of the scope of this study.

**Reviewer #2 (Recommendations For The Authors):**
Figure 1: This figure provides a description of the workflow. To make it more relevant for the paper, the authors could add relevant numbers as the workflow proceeds.(a) For example, how many memory B cells were sorted, how many supernatants were positive, and then how many mAbs were produced? These numbers can be attached to the relevant images in the workflow.

We modified the figure to include the numbers.

(b) For the "Supernatant screening via gamete extract ELISA", please change to "Supernatant screening via gamete/gametocyte extract ELISA".

We modified the statement as suggested.

Line 155: The manuscript states that 84 wells reacted with gamete/gametocyte lysate. The following sentence states that "Out of the 21 supernatants that were positive...". Can the authors provide the summary of data for all 84 wells or why focus on only 21 supernatants?

We screened all supernatants against gamete lysate, and only a subset against gametocyte lysate. In total, we found 84 positive supernatants that were reactive to at least one of the two lysates. 21 of those 84 positive were screened against both lysates. We have modified the text to clarify the numbers:

'After activation, single cell culture supernatants potentially containing secreted IgGs were screened in a high-throughput 384-well ELISA for their reactivity against a crude Pf gamete lysate (Fig. S1B). A subset of supernatants was also screened against gametocyte lysate (S1C). In total, supernatants from 84 wells reacted with gamete and/or gametocyte lysate proteins, representing 5.6% of the total memory B cells. Of the 21 supernatants that were screened against both gamete and gametocyte lysates, six recognized both, while nine appeared to recognize exclusively gamete proteins, and six exclusively gametocyte proteins.'

Please note that all 84 positive wells were taken forward for B cell sequencing and cloning.

Line 171: SIFA is introduced for the first time and should be completely spelled out.

We have corrected this.

Figure 2:(a) In Figure 2A, can you change the column title from "% pos KO GMT" to "% pos Pfs48/45 KO GMT"?

We have changed the column titles.

(b) In Figure 2B, the SMFA results have been converted to %TRA. Can the authors please provide the raw data for the oocyst counts and number of mosquitoes infected in Supplementary Materials?

We have added oocyst count data in Table S2, to which we refer in the figure legend.

(c) For Figure 2F, the authors do have other domains to Pfs230 as described in Inklaar et al, NPJ Vaccines 2023. An ELISA/Western to the other domains could identify the binding site for B2C10L, though we appreciate this is not the central result of this manuscript.We thank the reviewer for this suggestion. We are indeed planning to identify the target domain of B2C10L using the previously described fragments, but agree with the reviewer that this not the focus of the current manuscript and decided to therefore not include it in the current report.Line 116: The word sporozoites appears in subscript and should be corrected to be normal text.

We have corrected this.

Line 216: Typo "B1E11K"

We have corrected this.

Materials and Methods:(a) PBMC sampling: Please add the ethics approval codes in this section.

Donor A visited the hospital with a clinical malaria infection and provided informed consent for collection of PBMCs. We have modified the method section to clarify this.

'Donor A had lived in Central Africa for approximately 30 years and reported multiple malaria infections during that period. At the time of sampling PBMCs, Donor A had recently returned to the Netherlands and visited the hospital with a clinical malaria infection. After providing informed consent, PBMCs were collected, but gametocyte prevalence and density were not recorded.'

(b) Gamete/Gametocyte extract ELISA: Can the authors please provide the concentration of antibodies used for the positive and negative controls (TB31F, 2544, and 399)

We have added the concentrations for these mAbs in the methods section.

Recombinant Pfs48/45 and Pfs230 ELISA: Please state the concentration or molarity used for the coating of recombinant Pfs48/45 and Pfs230CMB.

We have added the concentrations, i.e. 0.5 µg/mL, to the methods section.

Western Blotting: The protocol states that DTT was added to gametocyte extracts (Line 594), but Western Blots in Figures 2 and 3 were performed in non-reducing conditions. Please confirm whether DTT was added or not.

Thank you for noting this. We did not use DTT for the western blots and have removed this line from the methods section.

**Reviewer #3 (Recommendations For The Authors):**
Below are a few minor comments to help improve the manuscript.(1) In Figure 4E, are the BLI data fit to a 1:1 binding model? The fits seem a bit off, and from ITC and X-ray studies it is known that 2 Fabs bind 1 peptide. The second Fab should presumably have higher affinity than the first Fab since the second Fab will make interactions with both the peptide and the first Fab. It may be better to fit the BLI data to a 2:1 binding model.

The 2:1 (heterogeneous ligand) model assumes that there are two different independent binding sites. However, the second binding event described is dependent on the first binding event and thus this model also does not accurately reflect the system. Given that there is not an ideal model to fit, we instead are careful about the language used in the main text to describe these results. Additionally, we also include a sentence to the results section to ensure that the proper findings/interpretations are highlighted: '…our data all trend toward the same finding in which a stronger binding affinity is observed toward the longer RESA P2 (16AA) peptide.'

(2) The sidechain interactions shown in Figures 5C and D could probably be improved. The individual residues are just 'floating' in space, causing them to lack context and orientation.

We added two panels (Fig. 5C-D) showcasing the binding interface that shows the peptide numbering in the context of the overall complex. We hope that this will help orient the reader.

(3) The percentage of Ramachandran outliers should be listed in Table 2. Presumably, the value is 0.2%, but this is omitted in the current table.

Table 2 has been modified to include the requested information explicitly.